# Neural Trees for Learning on Graphs

**Rajat Talak, Siyi Hu, Lisa Peng, and Luca Carlone**
Laboratory of Information and Decision Systems (LIDS)
Massachusetts Institute of Technology
{talak, siyi, lisapeng, lcarlone}@mit.edu

## Abstract

Graph Neural Networks (GNNs) have emerged as a flexible and powerful approach for learning over graphs. Despite this success, existing GNNs are constrained by their local message-passing architecture and are provably limited in their expressive power. In this work, we propose a new GNN architecture – the *Neural Tree*. The neural tree architecture does not perform message passing on the input graph, but on a tree-structured graph, called the *H-tree*, that is constructed from the input graph. Nodes in the H-tree correspond to subgraphs in the input graph, and they are reorganized in a hierarchical manner such that the parent of a node in the H-tree always corresponds to a larger subgraph in the input graph. We show that the neural tree architecture can approximate any smooth probability distribution function over an undirected graph. We also prove that the number of parameters needed to achieve an $\epsilon$-approximation of the distribution function is exponential in the treewidth of the input graph, but linear in its size. We prove that any continuous $\mathcal{G}$-invariant/equivariant function can be approximated by a nonlinear combination of such probability distribution functions over $\mathcal{G}$. We apply the neural tree to semi-supervised node classification in 3D scene graphs, and show that these theoretical properties translate into significant gains in prediction accuracy, over the more traditional GNN architectures. We also show the applicability of the neural tree architecture to citation networks with large treewidth, by using a graph sub-sampling technique.

## 1 Introduction

Graph-structured learning problems arise in several disciplines, including biology (*e.g.,* molecule classification [15]), computer vision (*e.g.,* action recognition [19], image classification [48], shape and pose estimation [30]), computer graphics (*e.g.,* mesh and point cloud classification and segmentation [20, 35, 41]), and social networks (*e.g.,* fake news detection [42]), among others [9]. In this landscape, *Graph Neural Networks* (GNN) have gained popularity as a flexible and effective approach for regression and classification over graphs.

Despite this growing research interest, recent work has pointed out several limitations of existing GNN architectures [56, 43, 38, 8]. Local message passing GNNs are no more expressive than the Weisfeiler-Lehman (WL) graph isomorphism test [56], neither can they serve as universal approximators to all $\mathcal{G}$-invariant (equivariant) functions, *i.e.,* functions defined over a graph $\mathcal{G}$ that remain unchanged by (or commute with) node permutation. The work [12] proves an equivalence between the ability to do graph isomorphism testing and the ability to approximate any $\mathcal{G}$-invariant function.

Various GNN architectures have been proposed, that go beyond local message passing or use tensor representations, in order to improve expressivity. Graph isomorphism testing, $\mathcal{G}$-invariant/equivariant function approximation, and the generalized $k$-order WL ($k$-WL) tests have served as end objectives and guided recent progress of this inquiry. For example, $k$-order linear GNN [39] and $k$-order folklore GNN [38] have expressive powers equivalent to $k$-WL and $(k + 1)$-WL test, respectively [3]. While

these architectures can theoretically approximate any $\mathcal{G}$-invariant function (as $k \rightarrow \infty$), they use $k$-order tensors for representations, rendering them impractical for any $k > 3$.

There is a need for a new way to look at constructing GNN architectures. With better end objectives to guide theoretical progress. Such an attempt can result in new and expressive GNNs that are provably tractable – if not in general, at least in reasonably constrained settings.

A GNN, by its very definition, operates on graph structured data. The graph structure of the data determines inter-dependency between nodes and their features. Probabilistic graphical models present a reasonable and well-established way of articulating and working with such inter-dependencies in the data. Prior to the advent of neural networks, inference algorithms on such graphical models were successfully applied to many real-world problems. Therefore, we pose that a GNN architecture operating on a graph should have at least the expressive power of a probabilistic graphical model, *i.e.,* it should be able to approximate any distribution defined by a probabilistic graphical model.

This is not a trivial requirement as exact inference (akin to learning the distribution or its marginals) on a probabilistic graphical model, without any structural constraints on the input graph, is known to be an NP-hard problem [13]. Even approximate inference on a probabilistic graphical model is known to be NP-hard in general [46]. A common trick to perform exact inference, consists in constructing a *junction tree* for an input graph and performing message passing on the junction tree instead. In the junction tree, each node corresponds to a subset of nodes of the input graph. The junction tree algorithm remains tractable for graphs with bounded treewidth, while [11] shows that treewidth is the only structural parameter, bounding which, allows for tractable inference on graphical models.

**Contribution.** We first define the notion of $\mathcal{G}$-compatible function and argue that approximating $\mathcal{G}$-compatible functions is equivalent to approximating any probability distribution on a probabilistic graphical model (Section 4); we also show that $\mathcal{G}$-invariant/equivariant functions considered in related work can be approximated using a nonlinear combination of $\mathcal{G}$-compatible functions.

We then propose a novel GNN architecture – the *Neural Tree* – that can approximate any $\mathcal{G}$-compatible function (Section 5). Neural trees do not perform message passing on the input graph, but on a tree-structured graph, called the *H-tree*, that is constructed from the input graph. Each node in the H-tree corresponds to a subgraph of the input graph. These subgraphs are arranged hierarchically in the H-tree such that the parent of a node in the H-tree always corresponds to a larger subgraph in the input graph. The leaf nodes in the H-tree correspond to singleton subsets (*i.e.,* individual nodes) of the input graph. The H-tree is constructed by recursively computing tree decompositions of the input graph and its subgraphs, and attaching them to one another to form a hierarchy. Neural message passing on the H-tree generates representations for all the nodes and important subgraphs of the input graph.

We next prove that the neural tree architecture can approximate any smooth $\mathcal{G}$-compatible function defined over a given undirected graph (Section 6). We also bound the number of parameters required by a neural tree architecture to obtain an $\epsilon$-approximation of an arbitrary (smooth) $\mathcal{G}$-compatible function. We show that the number of parameters increases exponentially in the treewidth of the input graph, but only linearly in the input graphs size. Thus, for graphs with bounded treewidth, the neural tree can tractably approximate any smooth distribution function.

We apply the neural tree architecture for semi-supervised node classification in 3D scene graphs and citation networks (Section 7). Our experiments on 3D scene graphs demonstrate that neural trees outperform standard, local message passing GNNs, by a large margin. Citation networks on the other hand, typically have large treewidth; therefore we make use of a recently proposed bounded treewidth graph sub-sampling algorithm [62], that sub-samples the input graph (*i.e.,* removes edges) to reduce its treewidth to a specified number. We show that applying the neural tree architecture in conjunction with such sub-sampling algorithm makes our architecture scalable to large graphs while still preserving its advantage over traditional architectures. Our code is publically available at `https://github.com/MIT-SPARK/neural_tree`

## 2 Related Work

**Expressive Power of Graph Neural Networks.** Since the seminal works [18, 49], various GNN architectures have been proposed including Graph Convolutional Networks (GCN) [29, 21, 14, 29, 9], Message Passing Neural Networks (MPNN) [17], GraphSAGE [33], Graph Attention Networks

(GAT) [53, 34, 10], message passing GNN [17]. Limited expressive power of these standard GNNs has been a major concern. For instance, it is known that local message passing GNNs can neither distinguish between non-isomorphic graphs (provably worse than the 1-Weisfeiler-Lehman (WL) test) [56, 43], nor can they compute even simple graph properties [16].

Many GNN architectures have been proposed to overcome this expressivity bottleneck. Graph substructure network is proposed in [8] and is shown to be more powerful than the 1-WL test. $k$-order GNNs, in which message passing is performed among a subset of nodes in the input graph, is shown to have expressive power equivalent to the generalized $k$-WL test [43, 38]. It is generally understood that to improve the expressivity of GNNs one has to extract features corresponding to important subgraphs, and operate on them. A hierarchical architecture that pools a representation vector from a subset of nodes, at each layer, is proposed in [61]. A junction-tree based message passing GNN is proposed for molecular graph generation in [23].

Graph neural networks have been investigated as function approximators since the beginning. [50] introduces the notion of *unfolding equivalence* and derives a universal approximation result for graph neural networks. Recent research in developing expressive GNN architectures has been towards approximating graph invariant/equivariant functions [40, 39, 38, 27, 47]. While, invariance and equivariance are desirable properties, the problem of designing GNNs that are universal approximators of $\mathcal{G}$-invariant/equivariant functions has been difficult. For instance, the $k$-order GNNs [39, 38] can provably approximate any graph invariant function, but only as $k \to \infty$, rendering them impractical [3]. An equivalence between designing GNN architectures to approximate graph invariant functions and graph isomorphism testing is shown in [12]. The generalization power of GNNs has also been investigated in [51, 16, 57, 58].

**Scene Graphs.** Scene graphs are a popular model to abstract information in images or model 3D environments. 2D scene graphs have been used in image retrieval [24], caption generation [26, 1], visual question answering [44, 31], and relationship detection [37]. GNNs are a popular tool for joint object labels and/or relationship inference on scene graphs [55, 36, 59, 63]. Recently, there has been a growing interest towards *3D* scene graphs, which are constructed from 3D data, such as meshes [2], point clouds [54], or raw sensor data [28, 45]. GNNs have been very recently applied to 3D scene graphs for scene layout prediction [54] or object search [32].

## 3   Problem Statement and Preliminaries

In this section, we state the node classification problem and review standard graph neural networks.

**Problem.** We focus on the standard problem of *semi-supervised node classification* [29]. We are given a graph $\mathcal{G} = (\mathcal{V}, \mathcal{E})$ along with node features $\boldsymbol{X} = (\boldsymbol{x}_v)_{v \in \mathcal{V}}$; where $\boldsymbol{x}_v$ denotes the node feature of node $v \in \mathcal{V}$. The graph is not necessarily connected. A subset of nodes $\mathcal{A} \subset \mathcal{V}$ in $\mathcal{G}$ are labeled, *i.e.,* $\{z_v \in \mathcal{L} \mid v \in \mathcal{A}\}$ is given; here $z_v$ denotes the label for node $v$ and $\mathcal{L}$ the finite set of label classes. We need to design a model to predict the labels of all the unlabeled nodes $v \in \mathcal{V} \backslash \mathcal{A}$.

**Graph Neural Networks (GNN).** Various GNN architectures have been successfully applied to solve the node classification problem [33, 29, 53, 56, 23]. Standard GNN architectures construct representation vectors for each node in $\mathcal{G}$ by iteratively aggregating representation vectors of its neighboring nodes. At iteration $t$, the representation vector of node $v$ is generated as follows:

$$h_v^t = \text{AGG}_t \left( h_v^{t-1}, \left\{ \left( h_u^{t-1}, \kappa_{u,v} \right) \mid u \in \mathcal{N}_{\mathcal{G}}(v) \right\} \right), \tag{1}$$

with $h_v^0 \triangleq \boldsymbol{x}_v, \forall v \in \mathcal{V}$; where $\mathcal{N}_{\mathcal{G}}(v)$ denotes the set of neighbors of node $v$ in graph $\mathcal{G}$ and the aggregation function $\text{AGG}_t$ can depend on the trainable edge parameters $\kappa_{u,v}$. This process of sharing and aggregating representation vectors among neighboring nodes in $\mathcal{G}$ is often called *message passing*. This procedure runs for a fixed number of iterations $T$. The node labels are then generated from the representation vectors at the final iteration $T$. Node labels are extracted as

$$y_v = \text{READ}(h_v^T), \tag{2}$$

for all $v \in \mathcal{V}$. The functions $\text{AGG}_t$ and READ are modeled as single or multi-layer perceptrons.

# 4 Graph Compatible Functions

We start by defining a class of $\mathcal{G}$-compatible functions. $\mathcal{G}$-compatible functions allow us to establish connections with probabilistic graphical models and the $\mathcal{G}$-invaraint/equivariant functions.

**Definition 1 ($\mathcal{G}$-compatible functions)** *We say that a function $f : (\times_{v \in \mathcal{V}} \mathbb{X}_v, \mathcal{G}) \to \mathbb{R}$ is compatible with graph $\mathcal{G}$ or $\mathcal{G}$-compatible if it can be factorized as*

$$f(\boldsymbol{X}) = \sum_{C \in \mathcal{C}(\mathcal{G})} \theta_C(\boldsymbol{x}_C), \tag{3}$$

*where $\mathcal{C}(\mathcal{G})$ denotes the collection of all maximal cliques in $\mathcal{G}$ and $\theta_C$ is some function that maps $\times_{v \in C} \mathbb{X}_v$ (the set of node features in the clique $C$) to a real number.*

Compatible functions arise in probabilistic graphical models; for instance, the logarithm of a joint probability distribution is a compatible function (see supplementary material for more examples on how such functions arise in inference on graphical models).

**Relation with Invariant/Equivariant Functions.** A graph invariant function requires that the function output remains invariant to node permutation, whereas a graph equivariant function outputs a vector (or a tensor in general) which is required to commute with any permutation applied to the input graph nodes. While graph invariance is a desirable property for graph classification problems, graph equivariance is desirable in node classification problems.

We now show that any continuous $\mathcal{G}$-invariant or $\mathcal{G}$-equivariant function can be written as a finite sum of $\mathcal{G}$-compatible functions, each composed with a specific nonlinear function. The precise definitions of $\mathcal{G}$-invariant and $\mathcal{G}$-equivariant functions and the proof is given in the supplementary material. .

**Theorem 2 (Invariance/Equivariance)** *For any continuous $\mathcal{G}$-invariant function $h : \mathbb{X} \to \mathbb{R}$ and an $\epsilon > 0$ there exists an integer $M \geq 1$ and a collection of $M$ continuous $\mathcal{G}$-compatible functions $\{f^i\}_{i=1}^M$ such that*

$$\sup_{\boldsymbol{X} \in \mathbb{X}} \left| h(\boldsymbol{X}) - \sum_{i=1}^M \phi\left(f^i(\boldsymbol{X})\right) \right| < \epsilon, \tag{4}$$

*where $\phi : \mathbb{R} \to \mathbb{R}$ is a nonlinear function. For any continuous $\mathcal{G}$-equivariant function $h : \mathbb{X} \to \mathbb{R}^n$ and an $\epsilon > 0$, (4) holds for each of its component $h_l(\boldsymbol{X})$.*

The result shows that a GNN architecture that can approximate any $\mathcal{G}$-compatible function will also be able to approximate any graph invariant and equivariant functions. In the next section, we describe the neural tree architecture, which can approximate any (smooth) $\mathcal{G}$-compatible function.

# 5 Neural Tree Architecture

The key idea behind the neural trees architecture is to construct a tree-structured graph from the input graph and perform message passing on the resulting tree instead of the input graph.

In the following, we first review the notion of tree decomposition (Section 5.1). We then show how to construct a H-tree for a graph, by successively applying tree decomposition on a given graph $\mathcal{G}$ and its subgraphs (Section 5.2). Finally, we discuss the proposed neural tree architecture for node classification, which performs neural message passing on the H-tree (Section 5.3).

In Section 6, we show that the tree structure enables the derivation of strong approximation results by which a neural tree can approximate any (smooth) $\mathcal{G}$-compatible function.

## 5.1 Tree Decomposition

For a graph $\mathcal{G}$, a *tree decomposition* is a tuple $(\mathcal{T}, \mathcal{B})$ where $\mathcal{T}$ is a tree graph and $\mathcal{B} = \{B_\tau\}_{\tau \in \mathcal{V}(\mathcal{T})}$ is a family of *bags*, where $B_\tau \subset \mathcal{V}(\mathcal{G})$ for every tree node $\tau \in \mathcal{V}(\mathcal{T})$, such that the tuple $(\mathcal{T}, \mathcal{B})$ satisfies the following two properties:

(1) *Connectedness:* for every graph node $v \in \mathcal{V}(\mathcal{G})$, the subgraph of $\mathcal{T}$ induced by tree nodes $\tau$ whose bag contains node $v$, is connected, *i.e.,* $\mathcal{T}_v \triangleq \mathcal{T}[\{\tau \in \mathcal{V}(\mathcal{T}) \mid v \in B_\tau\}]$ is a connected subgraph of $\mathcal{T}$ for every $v \in \mathcal{V}(\mathcal{G})$.

(2) *Covering:* for every edge $\{u, v\}$ in $\mathcal{G}$ there exists a node $\tau \in \mathcal{V}(\mathcal{T})$ such that $u, v \in B_\tau$.

The simplest tree decomposition of any graph $\mathcal{G}$ is a tree with a single node, whose bag contains all the nodes in $\mathcal{G}$. However, in practical applications, it is desirable to obtain decompositions where the size of the largest bag is small. This is captured by the notion of *treewidth*. The treewidth of a tree decomposition $(\mathcal{T}, \mathcal{B})$ is defined as the size of the largest bag minus one:

$$\text{tw}\left[(\mathcal{T}, \mathcal{B})\right] \triangleq \max_{\tau \in \mathcal{V}(\mathcal{T})} |B_\tau| - 1. \tag{5}$$

The treewidth of a graph $\mathcal{G}$ is defined as the minimum treewidth that can be achieved among all tree decompositions of $\mathcal{G}$. While finding a tree decomposition with minimum treewidth is NP-hard, many algorithms exist that generate tree decompositions with small enough treewidth [4, 5, 52, 6, 7].

We use $(\mathcal{T}, \mathcal{B}) = \text{tree-decomposition}(\mathcal{G})$ to denote a generic tree decomposition of a graph $\mathcal{G}$. One of the most popular tree decompositions is the junction tree decomposition, which was introduced in [22]. We denote it by $(\mathcal{T}, \mathcal{B}) = \text{junction-tree}(\mathcal{G})$ and describe it's construction in the supplementary material for completeness.

## 5.2 The H-tree

We first define H-tree for a complete graph. Let $\mathcal{S}_n$ denote a star graph with $n$ leaf nodes and one root.

**Definition 3 (Complete graph)**
*For a complete graph $\mathcal{G}$ with $n$ nodes, the H-tree is a star graph, i.e., $\mathcal{J}_\mathcal{G} = \mathcal{S}_n$, where the root node (in $\mathcal{J}_\mathcal{G}$) represents the single maximal clique in $\mathcal{G}$ and each of the leaf nodes in $\mathcal{S}_n$ corresponds to a node in $\mathcal{G}$.*

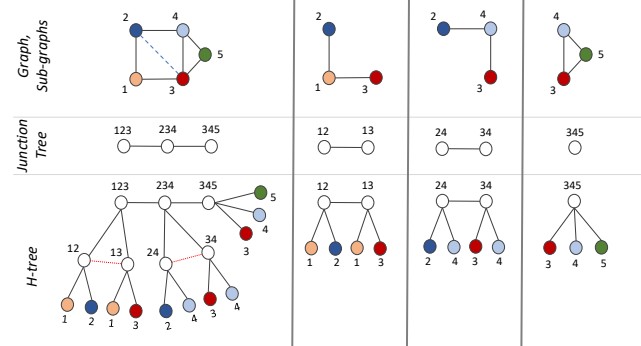

Figure 1: Generating H-trees for graph $\mathcal{G}$ and its subgraphs.

The H-tree for a complete graph of three nodes is shown in Fig. 1, rightmost column. In it, the unique clique in the graph, which contains nodes $\{3, 4, 5\}$, is labeled as $C = (345)$. For the sake of clarity, we always enlist the set of root nodes when defining an H-tree. Therefore, an H-tree of a graph $\mathcal{G}$ is given by a tuple $(\mathcal{J}_\mathcal{G}, R)$, where $\mathcal{J}_\mathcal{G}$ is a tree graph and $R$ is the set of *root* nodes.

---

**Algorithm 1: H-tree**

**input** : Graph $\mathcal{G}$
**output** : H-tree $(\mathcal{J}_\mathcal{G}, R)$

1 $(\mathcal{T}, \mathcal{B}) \leftarrow \text{tree-decomposition}(\mathcal{G})$
2 $\mathcal{J} \leftarrow \mathcal{T}$
3 $R \leftarrow \mathcal{V}(\mathcal{T})$
4 **for** each bag $B$ in $\mathcal{B}$ **do**
5     **if** $\mathcal{G}[B]$ is a complete graph **then**
6         Update $\mathcal{J}$:
7         $\mathcal{V}(\mathcal{J}) \leftarrow \mathcal{V}(\mathcal{J}) \cup B$
8         $\mathcal{E}(\mathcal{J}) \leftarrow \mathcal{E} \cup \{\{\tau(B), b\}\}_{b \in B}$
9     **else**
10         $(\mathcal{J}', R') \leftarrow \text{H-tree}(\mathcal{G}[B])$
11         Update $\mathcal{J}$:
12         $\mathcal{E}(\mathcal{J}') \leftarrow \mathcal{E}(\mathcal{J}') \backslash \mathcal{E}(\mathcal{J}'[R'])$
13         $\mathcal{J} \leftarrow \mathcal{J} \cup \mathcal{J}'$
14         $\mathcal{E}(J) \leftarrow \mathcal{E}(J) \cup \{\{\tau(B), r\}\}_{r \in R'}$
15     **end**
16     $\mathcal{J}_G \leftarrow \mathcal{J}$
17     return $(\mathcal{J}_\mathcal{G}, \text{R})$
18 **end**

---

The H-tree is computed by recursively applying tree decomposition on the input graph and the subgraphs obtained in tree decomposition. For instance, if $(\mathcal{T}, \mathcal{B})$ is a tree decomposition of the input graph $\mathcal{G}$, then we recursively apply tree decomposition to each subgraph $\mathcal{G}[B]$ (of $\mathcal{G}$) for each $B \in \mathcal{B}$. The final H-tree $\mathcal{J}_\mathcal{G}$ is computed by connecting all the obtained tree decomposition as a hierarchy. The set of root nodes $R$ are the nodes in $\mathcal{J}_\mathcal{G}$ corresponding to the original tree decomposition $(\mathcal{T}, \mathcal{B})$ of the graph. This process is illustrated in Figure 1 and the algorithm described in Algorithm 1.

We now describe the algorithm in more detail. Algorithm 1 takes an undirected graph $\mathcal{G}$ and outputs a H-tree $\mathcal{J}_\mathcal{G}$ with a set of root nodes $R$. Let $(\mathcal{T}, \mathcal{B})$ denote a tree decomposition of graph $\mathcal{G}$ (line 1). The H-tree $\mathcal{J}$ is initialized to $\mathcal{J} = \mathcal{T}$ (line 2) and the set of root nodes equals the root nodes of this tree, namely $R = \mathcal{V}(\mathcal{T})$ (line 3). For $B \in \mathcal{B}$, let $\tau(B)$ denote the node corresponding to bag $B$ in $\mathcal{J}$. Then for each bag $B \in \mathcal{B}$ we construct a H-tree of the induced subgraph $\mathcal{G}[B]$ (lines 4-17).

If $\mathcal{G}[B]$ is not complete, we attach its root nodes to $\tau(B)$ (lines 9-15). Specifically, if $(\mathcal{J}', R')$ denotes the H-tree for the induced subgraph $\mathcal{G}[B]$, then we attach the graph $\mathcal{J}'$ to $\mathcal{J}$ by linking all root nodes of $\mathcal{J}'$,

namely $R'$, to the node $\tau(B)$ (lines 13-14). To avoid cycles, we also remove edges between the root nodes $R'$ in $\mathcal{J}'$ (line 12).

If the induced subgraph $\mathcal{G}[B]$ is complete, then from Definition 3 we know that its H-tree is a star graph with a single clique node, call it $C$. In this case, we attach the star graph to $\tau(B)$ by merging two nodes – $C$ and $\tau(B)$ – into one. This avoids an unnecessary edge $(\tau(B), C)$ in the H-tree.

**Example.** Figure 1 shows the construction of a H-tree for a graph with 5 nodes and 6 edges. Here, we have used the `junction-tree` algorithm to perform tree decomposition. The first column shows the graph $\mathcal{G}$ and its junction trees, which has three nodes corresponding to the three cliques in the chordal graph $\mathcal{G}_c$ (which in this case consists in adding the dashed blue line in Figure 1; see supplementary material for details). The remaining columns show the three subgraphs of $\mathcal{G}$ corresponding to each of the three maximal cliques in $\mathcal{G}_c$, along with their junction trees and H-trees. The H-tree of each of these subgraphs is then attached to the junction tree of $\mathcal{G}$ to get the required H-tree for $\mathcal{G}$. The H-tree for graph $\mathcal{G}$ is shown in the last row of the first column in Figure 1. Also illustrated are the two edges deleted (in red) when merging the two H-trees of the subgraphs to the junction tree of $\mathcal{G}$.

**Remark 4 (Leaves and features)** *Each node in the H-tree (of a graph $\mathcal{G}$) corresponds to a subset of nodes in graph $\mathcal{G}$. Every leaf node $l$ in $\mathcal{J}_\mathcal{G}$ corresponds to exactly one node $v$ in $\mathcal{G}$. We denote this node by $\kappa(l)$ for every leaf node $l$ of $\mathcal{J}_\mathcal{G}$. In the construction of the H-tree, we also assign the node input feature $\boldsymbol{x}_v$ to every node $l$ in $\mathcal{J}_\mathcal{G}$ for which $\kappa(l) = v$. Note that multiple leaf nodes may correspond to a single node $v$ in the graph $\mathcal{G}$, i.e., we can have $\kappa(l) = v$ for many leaf nodes $l$ in $\mathcal{J}_\mathcal{G}$. Fig. 1 illustrates the input node features by node coloring.*

## 5.3 Message Passing on the H-tree

Given a graph $\mathcal{G}$ with input node features, we construct a H-tree $(\mathcal{J}_\mathcal{G}, R)$ and perform message passing on $\mathcal{J}_\mathcal{G}$. We call this the *neural tree* architecture. Representation vectors are generated for each node in the H-tree $\mathcal{J}_\mathcal{G}$ by aggregating representation vectors of neighboring nodes in $\mathcal{J}_\mathcal{G}$. The message passing starts with $h_l^0 = \boldsymbol{x}_{\kappa(l)}$ for all leaf nodes $l$ in $\mathcal{J}_\mathcal{G}$ and $h_u^0 = \boldsymbol{0}$ for non-leaf nodes $u$ in $\mathcal{J}_\mathcal{G}$. These representation vectors are then updated as

$$\boldsymbol{h}_u^t = \text{AGG}_t\left(\boldsymbol{h}_u^{t-1}, \{\left(\boldsymbol{h}_w^{t-1}, \kappa_{w,u}\right) \mid w \in \mathcal{N}_{\mathcal{J}_\mathcal{G}}(u)\}\right), \tag{6}$$

for each iteration $t \in \{1, 2, \ldots T\}$. The aggregation function $\text{AGG}_t$ can be modeled in numerous ways. Many of the message passing GNN architectures in the literature, such as GCN [29], Graph-SAGE [33], GIN [56], GAT [53, 34], can be used to perform message passing on $\mathcal{J}_\mathcal{G}$. The message passing in (6), using edge weights, can also be made to distinguish between edges connecting to roots and children in the H-tree. After $T$ iterations of message passing, we extract the label $y_v$ for node $v \in \mathcal{G}$ by combining the representation vectors of leaf nodes $l$ of $\mathcal{J}_\mathcal{G}$, which correspond to node $v$ in $\mathcal{G}$, i.e., $v = \kappa(l)$:

$$y_v = \text{COMB}\left(\{\boldsymbol{h}_l^T \mid l \text{ leaf node in } \mathcal{J}_\mathcal{G} \text{ s.t. } \kappa(l) = v\}\right), \tag{7}$$

for every $v \in \mathcal{V}$, where $\boldsymbol{h}_l^T$ denotes the representation vector generated at leaf node $l$ in $\mathcal{J}_\mathcal{G}$ after $T$ iterations. COMB can be modeled by using any of the standard neural network models. In our experiments, we model COMB with a mean pooling function followed by a softmax.

**Remark 5 (Mutatis mutandis)** *The neural tree architecture is partly inspired by the junction tree algorithm [25]. The junction tree message passing algorithm can be described in three steps. First, the clique potentials are computed for all nodes in the junction tree $(\mathcal{T}, \mathcal{B})$ of $\mathcal{G}$. This is followed by message passing between nodes on $\mathcal{T}$, which updates the clique potentials, until convergence. Third, the marginals are computed for each node from the clique potentials. The proposed neural tree can emulate these three steps by message passing from leaf nodes to the root nodes in H-tree, message passing between the root nodes, and message passing back from the root nodes to the leaf nodes, respectively. [57] suggests that such algorithmic alignment of the neural architecture leads to better generalizability. We leave the question of generalization power to future work.*

**Remark 6 (Scalability and trade-offs)** *The proposed architecture requires constructing the H-tree for the graph $\mathcal{G}$, which involves computing a tree decomposition of $\mathcal{G}$. The time and space complexity of computing a tree decomposition of a graph $\mathcal{G}$ scales exponentially in the treewidth of $\mathcal{G}$. In many semi-supervised node classification problems, the treewidth of the input graph is too large to compute*

*a tree decomposition (eg. graphs arising in citation networks [60]). In such cases, to regain computational tractability, one can sub-sample the input graph (i.e., remove some edges in $\mathcal{G}$) to get a graph $\mathcal{G}_s$ with smaller treewidth, and then apply the neural tree architecture to this sub-sampled graph $\mathcal{G}_s$. [62] proposes one such graph sub-sampling algorithm, which for any given graph $\mathcal{G}$ and a treewidth bound $k$, efficiently generates the sub-sampled graph $\mathcal{G}_s$ and its tree decomposition. The complexity of this algorithm is $\mathcal{O}(|\mathcal{E}(\mathcal{G})|(k^2 + |\mathcal{V}(\mathcal{G})|))$. This addition to the neural tree architecture makes it scalable to large graphs (see Section 7.2).*

## 6 Expressive Power of Neural Trees

We now show that neural trees can learn any graph-compatible function provided it is smooth enough. All proofs are given in the supplementary material.

For simplicity, let the input node features and the representation vectors be real numbers, *i.e.*, $\boldsymbol{x}_v \in \mathbb{R}$ and $\boldsymbol{h}_u^t \in \mathbb{R}$ for all $v \in \mathcal{V}(\mathcal{G})$ and nodes $u \in \mathcal{J}_\mathcal{G}$. Let us implement the aggregation function $\text{AGG}_t$ in (6) as a shallow network:

$$\boldsymbol{h}_u^t = \text{AGG}_t\left(\boldsymbol{h}_u^{t-1}, \{(\boldsymbol{h}_w^{t-1}, \kappa_{w,u}) \mid w \in \mathcal{N}_{\mathcal{J}_\mathcal{G}}(u)\}\right) = \text{ReLU}\left(\sum_{k=1}^{N_u} a_{u,t}^k \langle \boldsymbol{w}_{u,t}^k, \boldsymbol{h}_{\bar{\mathcal{N}}(u)}^{t-1}\rangle + b_{u,t}^k\right), \tag{8}$$

where $\bar{\mathcal{N}}(u) = \{u\} \cup \mathcal{N}_{\mathcal{J}_\mathcal{G}}(u)$ denotes the set containing node $u$ and its neighbors in the H-tree $\mathcal{J}_\mathcal{G}$, and $a_{u,t}^k, b_{u,t}^k, \boldsymbol{w}_{u,t}^k$, and $N_u$ are parameters.[1] The representation vectors $\boldsymbol{h}_u^t$ are initialized as discussed in Section 5.3. We fix a node $v_0$ in graph $\mathcal{G}$ and extract our output from $v_0$:

$$y_{v_0} = \text{COMB}\left(\{\boldsymbol{h}_l^T \mid l \text{ leaf node in } \mathcal{J}_\mathcal{G} \text{ s.t. } \kappa(l) = v_0\}\right), \tag{9}$$

where $T$ is the number of iterations. We also assume the COMB function to be a shallow network.

Let $N$ denote the total number of parameters used in the neural tree architecture. Consider the space of functions $g$ that map the input node features $\boldsymbol{X}$ to the output $y_{v_0}$ in (8)-(9):

$$\mathcal{F}(\mathcal{G}, N) = \left\{g : \boldsymbol{X} \to g(\boldsymbol{X}) = y_{v_0} \;\middle|\; \begin{array}{l} \text{For some } T > 0 \text{ where} \\ y_{v_0} \text{ is given by (8)-(9)} \end{array}\right\}.$$

We now show that any graph-compatible function – that is smooth enough – can be approximated by a function in $\mathcal{F}(\mathcal{G}, N)$ to an arbitrary precision.

**Theorem 7** *Let $f : [0,1]^n \to [0,1]$ be a function compatible with a graph $\mathcal{G}$ with $n$ nodes. Let each clique function $\theta_c$ in $f$ (see Definition 1) be 1-Lipschitz and be bounded to $[0,1]$. Then, for any $\epsilon > 0$, there exists a $g \in \mathcal{F}(\mathcal{G}, N)$ such that $||f - g||_\infty < \epsilon$, while the number of parameters $N$ is bounded by*

$$N = \mathcal{O}\left(\sum_{u \in \mathcal{V}(\mathcal{J}_\mathcal{G})}(d_u - 1)\left(\frac{\epsilon}{d_u - 1}\right)^{-(d_u - 1)}\right), \tag{10}$$

*where $d_u$ denotes the degree of node $u$ in $\mathcal{J}_\mathcal{G}$, and the summation is over all the non-leaf nodes in $\mathcal{J}_\mathcal{G}$.*

Theorem 7 assumes the domain and range of the compatible function $f$ and the clique functions $\theta_c$ to be bounded between $[0,1]$. We remark here that the result, and the proof, can be extended to any bounded $f$ and $\theta_c$, over bounded domains.

We next develop the bound in Theorem 7 to expose the dependence of the number of parameters $N$ on the treewidth of the tree decomposition of the graph.

**Corollary 8** *The number of parameters $N$ in Theorem 7 is upper-bounded by*

$$N = \mathcal{O}\left(n \times (tw\,[\mathcal{J}_\mathcal{G}] + 1)^{2tw[\mathcal{J}_\mathcal{G}]+3} \times \epsilon^{-(tw[\mathcal{J}_\mathcal{G}]+1)}\right),$$

*where $tw\,[\mathcal{J}_\mathcal{G}]$ denotes the treewidth of the tree-decomposition of $\mathcal{G}$, formed by the root nodes of $\mathcal{J}_\mathcal{G}$.*

---

[1]We assume a different $\text{AGG}_t$ function for each node $u$ at iteration $t$. This choice is more general than our architecture in Section 5.3. However, our results extend to the case where the $\text{AGG}_t$ function is the same across nodes $u$ in each iteration $t$.

**Remark 9 (Efficient approximations)** *Corollary 8 shows that the number of parameters needed to obtain an $\epsilon$-approximation with neural trees increases exponentially in only the treewidth of the tree-decomposition, and is linear in the number of nodes $n$ in the graph. Thus, for graphs with bounded treewidth, neural trees are able to approximate any graph-compatible function efficiently.*

**Remark 10 (Data efficiency)** *The value of $N$ also affects the data required to train the model: the larger the $N$, the more samples are required for training. In particular, Corollary 8 provides the reassuring result that if the training dataset contains graphs of small treewidth, then the amount of data required for training scales only linearly in the number of nodes $n$.*

# 7 Experiments

### 7.1 Node Classification in 3D Scene Graphs

We use the neural tree architecture for node classification on 3D scene graphs and show it outperforms the standard GNNs.

**Dataset.** We run semi-supervised node classification experiments on Stanford's 3D scene graph dataset [2]. The dataset includes 35 3D scene graphs with verified semantic labels, each containing building, room, and object nodes in a residential unit. Since there is only a single class of building nodes (residential), we remove the building node and obtain 482 room-object graphs where each graph contains a room and at least one object in that room as shown in Fig. 2. The resulting dataset has 482 room nodes with 15 semantic labels, and 2338 objects with 35 labels. Each object node is connected to the room node it

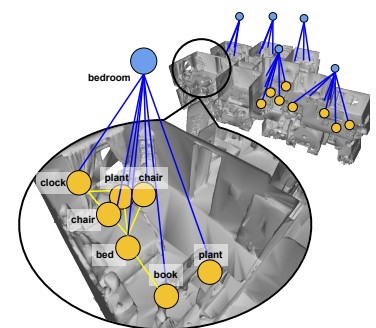

Figure 2: A room-object graph in a 3D scene graph.

belongs to. In addition we add 4920 edges to connect adjacent objects in the same room. We use the centroid and bounding box dimensions as features for each node.

**Approaches and Setup.** We implement the neural tree architecture with four different aggregation functions $\text{AGG}_t$ specified in: GCN [29], GraphSAGE [33], GAT [53], GIN [56]. We randomly select 10% of the nodes for validation and 20% for testing. The hyper-parameters of the two approaches are separately tuned based on the best validation accuracy, while using all 70% of the remaining nodes for training; see supplementary material for details.

**Results.** Table 1 compares the test accuracies (averaged over 100 runs) for the standard GNN architectures and the corresponding neural tree architecture, while using the same type of aggregation function. We see that the neural tree architecture always yields a better prediction model than the standard GNN, for a given aggregation function.

Table 1: Test Accuracy

| Model | Input graph | Neural Tree |
| --- | --- | --- |
| GCN | $40.88 \pm 2.28\,\%$ | $\mathbf{50.63} \pm 2.25\,\%$ |
| GraphSAGE | $59.54 \pm 1.35\,\%$ | $\mathbf{63.57} \pm 1.54\,\%$ |
| GAT | $46.56 \pm 2.21\,\%$ | $\mathbf{62.16} \pm 2.03\,\%$ |
| GIN | $49.25 \pm 1.15\,\%$ | $\mathbf{63.53} \pm 1.38\,\%$ |

To further analyze the proposed architecture, we carry out a series of experiments to see how the test accuracy varies as a function of the amount of training data and the number of message passing iterations $T$. For simplicity, we only show the neural tree that uses the GCN aggregation function in comparison with the standard GCN.

Figures 3 and 4 plot the test accuracy (averaged over 10 runs) as a function of the training data used and the number of iterations $T$. The test accuracy –for both the neural tree (NT+GCN) and

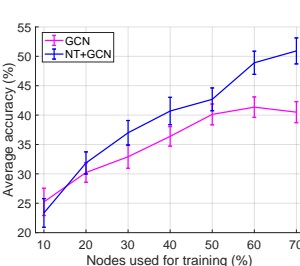

Figure 3: Accuracy vs. training data (% of labeled nodes).

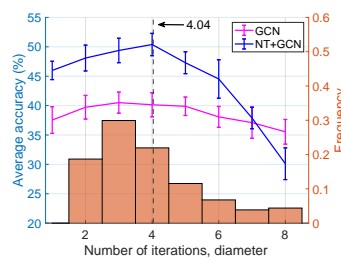

Figure 4: Accuracy vs. number of iterations and weighted diameter distribution.

GCN– increases with increasing training data, however, the increase is sharper for the neural tree architecture, eventually outperforming GCN.

This shows the higher expressive power of the proposed neural tree architecture. As with the number of iterations $T$, we see an optimal $T$ at which the test accuracy is maximized. This optimal $T$ is empirically close to the average diameter of the constructed H-trees, of all (room-object) scene graphs in the dataset. This is intuitive, as for the messages to propagate across the entire H-tree, $T$ would have to equal the diameter of the H-tree. See supplementary material for more details, where we also report the compute, train, and test time requirements for neural trees.

### 7.2 Node Classification in Citation Networks

Here we show that the neural tree architecture can scale to large citation network datasets by using the bounded treewidth graph sub-sampling proposed in [62]. We use the popular citation network datasets [60], where nodes are documents and undirected edges are citations. Each node has a class label representing the subject of the document. These graphs have high treewidth, and therefore, are first sub-sampled using the bounded treewidth graph sub-sampling algorithm in [62]. The neural tree is constructed on the sub-sampled graph.

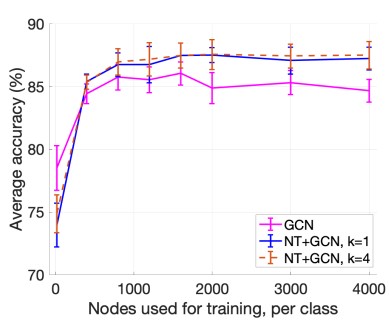

Figure 5: Accuracy vs. training data (per class).

In Figure 5 we plot test accuracy (averaged over 10 runs) as a function of training data, for GCN and NT+GCN, on the Pubmed dataset [60]. As in the scene graph case, we again see that the test accuracy improves with increasing training data, and eventually the neural tree architecture performs better than the standard GNN (GCN in this case). We also observe no significant change in performance of the NT+GCN, with increasing treewidth bound $k$ used for graph sub-sampling. This indicates that it is possible to retain the best possible performance, even after sub-sampling the input graph with a very low treewidth bound. We report the details and more results in the supplementary material.

## 8 Conclusion

We propose a novel graph neural network architecture – the *neural tree*. The neural tree performs message passing, not on the input graph, but over a constructed H-tree, which provides a tree-structured description of the original graph and its subgraphs. We show that the neural tree architecture can approximate any graph-compatible function, and that the number of parameters required to obtain a desired approximation grows linearly with the number of nodes and exponentially in the treewidth of the input graph. This renders the proposed architecture more parsimonious for large graphs with small treewidth.

Graph-compatible functions arise in probabilistic graphical models, hence the proposed architecture can approximate any probability distribution function defined on a graph. Furthermore, we show that a graph-compatible function can be used to approximate any smooth graph-invariant/equivariant functions studied in the literature. This suggests that the goal of approximating graph-compatible functions is a worthwhile pursuit towards the design of novel GNN architectures.

We use neural trees for node classification on 3D scene graph and citation network datasets, showing that the proposed architecture leads to more accurate predictions with increasing training data and is applicable even for large networks with high treewidth.

*Neural Tree* is a general purpose architecture and remains to be applied to other learning tasks such as graph representation learning and classification.

## 9 Societal Impact

**Research Community.** Many problems have been sought to be solved using graph neural networks. However, the relation between complexity of the underlying problem and parameter complexity of

the neural architecture used to solve it is not generally well investigated. Moreover, the expressivity of the graph neural network architecture, *i.e.,*, its ability to solve any instance of the problem is also not fully understood.

This work, we believe, is a step towards understanding these fundamental questions. In obtaining approximation guarantees for the proposed *Neural Tree* architecture, we bring out an interesting tangle between approximating graph compatible functions (which can be thought of as approximating exact inference over probabilistic graphical models), graph treewidth, and the parameter complexity of the Neural Tree. The parameter complexity obtained in Theorem 8 matches the problem complexity of exact inference on probabilistic graphical models [11].

We hope that this work will inspire other researchers to consider similar questions - for other problems and neural architectures - and investigate the relation between the problem complexity, parameter complexity, and the underlying graph properties - such as the graph treewidth.

**Community at Large.** The main thrust of this work is to develop a new graph neural network architecture that can approximate any graph compatible function. We show that the parameter complexity increases exponentially in the graph treewidth, and is of the same order as the complexity of exact inference on graphical models. This implies that when applying Neural Trees, graph treewidth is not only an important parameter, but the most important aspect in controlling the required memory and computation time.

This can be a limiting factor in deploying the Neural Tree architecture in cases where either the energy consumption or large graph treewidth is an issue. In the paper, however, we observe that using the Neural Tree architecture in conjunction with bounded treewidth subgraph sampling [62] provides a good approximation in such cases.

## Acknowledgement

This work was partially funded by the Office of Naval Research under the ONR RAIDER program (N00014-18-1-2828).

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
