# Neural Trees for Learning on Graphs - Supplementary Material

**Rajat Talak, Siyi Hu, Lisa Peng, and Luca Carlone**
Laboratory of Information and Decision Systems (LIDS)
Massachusetts Institute of Technology
{talak, siyi, lisapeng, lcarlone}@mit.edu

This document serves the following purposes:

- it discusses how graph-compatible functions naturally arise in probabilistic graphical models and inference problems (Appendix A);

- it reviews a standard procedure to obtain a junction tree decomposition of a graph (Appendix B);

- it proves the key technical results in the main paper, including Theorem 2 (proof in Appendix C), Theorem 7 (proof in Appendix D), and Corollary 8 (proof in Appendix E);

- it provides extra experimental results and implementation details (Appendix F and G).

Our code is publically available at `https://github.com/MIT-SPARK/neural_tree`

35th Conference on Neural Information Processing Systems (NeurIPS 2021).

# A   Simple Examples of $\mathcal{G}$-Compatible Functions

Compatible functions naturally arise when performing inference on probabilistic graphical models.

**Probabilistic Graphical Models.** The joint probability distribution of a probabilistic graphical model, on an undirected graph $\mathcal{G}$, is given by

$$p(\boldsymbol{X} \mid \mathcal{G}) = \prod_{C \in \mathcal{C}(\mathcal{G})} \psi_C(\boldsymbol{x}_C), \tag{A1}$$

where $\mathcal{C}(\mathcal{G})$ is the collection of maximal cliques in $\mathcal{G}$ and $\psi_C$ are some functions, often referred to as *clique potentials* [4, 6]. This can be written as

$$p(\boldsymbol{X} \mid \mathcal{G}) = \exp\{f(\boldsymbol{X})\}, \tag{A2}$$

where $f(\boldsymbol{X})$ is a $\mathcal{G}$-compatible function according to (3), in the main paper, with $\theta_C(\boldsymbol{x}_C) = \log \psi_C(\boldsymbol{x}_C)$. Thus, the ability to approximate any $\mathcal{G}$-compatible function is equivalent to the ability to approximate any distribution function of a probabilistic graphical model, on graph $\mathcal{G}$.

We now provide two examples where we have to learn graph compatible functions to compute maximum likelihood estimates over graphs.

**Graph Classification.** Given a graph $\mathcal{G}$ and its label $y \in \mathcal{L}$, suppose that the node features $\boldsymbol{X}$ are distributed according to a probabilistic graphical model on the undirected graph $\mathcal{G}$. This induces a natural correlation between the observed node features, which is dictated by the graph $\mathcal{G}$. Then, the conditional probability density $p(\boldsymbol{X}|y, \mathcal{G})$ of the node features $\boldsymbol{X}$, given label $y$ and graph $\mathcal{G}$, is given by

$$p(\boldsymbol{X}|y, \mathcal{G}) = \prod_{C \in \mathcal{C}(\mathcal{G})} \psi_C(x_C, y), \tag{A3}$$

where $\mathcal{C}(\mathcal{G})$ is the set of all maximal cliques in graph $\mathcal{G}$ and $\psi_C$ are the clique potentials. A maximum likelihood estimator for the graph labels will predict:

$$\hat{y} = \arg\max_{y \in \mathcal{L}} \ \log p(\boldsymbol{X}|y, \mathcal{G}) = \arg\max_{y \in \mathcal{L}} \sum_{C \in \mathcal{C}(\mathcal{G})} \log \psi_C(x_C, y), \tag{A4}$$

whose objective is a $\mathcal{G}$-compatible function. In practice, we do not know the functions $\psi_C$ or the conditional distribution $p(\boldsymbol{X}|y, \mathcal{G})$, and will have to learn from the data to make predictions given in (A4) feasible.

**Node Classification.** Given a graph $\mathcal{G}$ and node labels $\boldsymbol{y} = \{y_v\}_{v \in \mathcal{V}(\mathcal{G})}$, suppose the node features $\boldsymbol{X}$ be distributed according to a probabilistic graphical model on graph $\mathcal{G}$. We, therefore, have

$$p(\boldsymbol{X}|\boldsymbol{y}, \mathcal{G}) = \prod_{C \in \mathcal{C}(\mathcal{G})} \psi_C(\boldsymbol{x}_C, \boldsymbol{y}). \tag{A5}$$

A maximum likelihood estimator that estimates node labels $\boldsymbol{y}$ by observing the node features will predict:

$$\hat{\boldsymbol{y}} = \arg\max_{y_v \in \mathcal{L}} \ \log p(\boldsymbol{X}|\boldsymbol{y}, \mathcal{G}) = \arg\max_{y_v \in \mathcal{L}} \sum_{C \in \mathcal{C}(\mathcal{G})} \log \psi_C(\boldsymbol{x}_C, \boldsymbol{y}), \tag{A6}$$

whose objective is a $\mathcal{G}$-compatible function.

**Remark 1 (Applying to directed graphs)** *The proposed model can be used to approximate inference on the directed graphical models as well. Note that the joint distribution on a directed graphical model can also be described as a product of clique potentials [4, 6]. However, we would first convert the directed model into an undirected graphical model using the technique of moralization [4, 6]. The H-tree can then be constructed on this undirected, moralized graph.*

# B   Junction Tree Decomposition

This section reviews the *junction tree* algorithm, proposed in [3]. We denote by $(\mathcal{T}, \mathcal{B}) =$ `junction-tree`$(\mathcal{G})$ the algorithm that takes an arbitrary graph $\mathcal{G}$ and returns a junction tree decomposition $(\mathcal{T}, \mathcal{B})$ as described below.

In order to obtain a junction tree decomposition of a given undirected graph $\mathcal{G}$, the graph $\mathcal{G}$ is first triangulated. Triangulation is done by adding a chord between any two nodes in every cycle of length $4$ or more. This eliminates all the cycles of length $4$ or more in the graph $\mathcal{G}$ to produce a chordal graph $\mathcal{G}_c$. The collection of bags $\mathcal{B} = \{B_\tau\}_\tau$ in the junction tree is chosen as the set of all maximal cliques in the chordal graph $\mathcal{G}_c$. Then, an *intersection graph* $\mathcal{I}$ on $\mathcal{B}$ is built, which has a node for every bag in $\mathcal{B}$ and an edge between two bags $B_\tau$ and $B_\mu$ if they have a non-empty intersection, *i.e.,* $|B_\tau \cap B_\mu| \geq 1$. The weight of every link $\{\tau, \mu\}$ in the intersection graph $\mathcal{I}$ is set to $|B_\tau \cap B_\mu|$. Finally, the desired junction tree is obtained by extracting a maximum weight spanning tree on the weighted intersection graph $\mathcal{I}$. It is know that this extracted tree $\mathcal{T}$, with the bag $\mathcal{B}$, is a valid tree-decomposition of $\mathcal{G}$ that satisifes the connectedness and covering property.

The junction tree decomposition of a graph and its subgraphs is shown in Fig. 1.

# C Invariant and Equivariant Function Approximation

In this section, we prove Theorem 2. We first recall the definitions of $\mathcal{G}$-invariant and $\mathcal{G}$-equivariant functions. Let $\boldsymbol{X}^\sigma$ to denote the tuple $(\boldsymbol{x}_{\sigma(v)})_{v \in \mathcal{V}}$, where $\sigma$ is a permutation of nodes $\mathcal{V}$ in graph $\mathcal{G}$. Define $\mathcal{E}^\sigma = \{(\sigma(u), \sigma(v)) \mid (u,v) \in \mathcal{E}\}$, for edges $\mathcal{E}$ in graph $\mathcal{G}$, and note that $\mathcal{G}^\sigma = (\mathcal{V}, \mathcal{E}^\sigma)$ is a permutation of graph $\mathcal{G}$.

**Definition 2 ($\mathcal{G}$-invariant function)** *A function $h : (\mathbb{X}, \mathcal{G}) \to \mathbb{R}$ is invariant with respect to graph $\mathcal{G}$ or $\mathcal{G}$-invariant if*

$$h(\boldsymbol{X}^\sigma, \mathcal{G}^\sigma) = h(\boldsymbol{X}, \mathcal{G}), \tag{A7}$$

*for all permutations $\sigma$ on $\mathcal{V}(\mathcal{G})$.*

**Definition 3 ($\mathcal{G}$-equivariant function)** *A function $h : (\mathbb{X}, \mathcal{G}) \to \mathbb{R}^n$ is equivariant with respect to graph $\mathcal{G}$ or $\mathcal{G}$-equivariant if*

$$h(\boldsymbol{X}^\sigma, \mathcal{G}^\sigma) = h(\boldsymbol{X}, \mathcal{G})^\sigma, \tag{A8}$$

*for all permutations $\sigma$ on $\mathcal{V}(\mathcal{G})$, where for a $\boldsymbol{z} \in \mathbb{R}^n$, $\boldsymbol{z}^\sigma \in \mathbb{R}^n$ is such that $\boldsymbol{z}_i^\sigma = \boldsymbol{z}_{\sigma(i)}$ for all $i \in [n]$.*

Theorem 2 can restated, in more detail, as follows:

**Theorem 4** *The following statements hold true.*

1. *For any continuous $\mathcal{G}$-invariant function $h$ and a scalar $\epsilon > 0$ there exists an integer $M \geq 1$ and a collection of $M$ continuous $\mathcal{G}$-compatible functions $\{f^i\}_{i=1}^M$ such that*

$$\sup_{\boldsymbol{X} \in \mathbb{X}} \left| h(\boldsymbol{X}, \mathcal{G}) - \sum_{i=1}^M \phi\left(f^i(\boldsymbol{X}, \mathcal{G})\right) \right| < \epsilon, \tag{A9}$$

   *where $\phi : \mathbb{R} \to \mathbb{R}$ is some function.*

2. *For any continuous $\mathcal{G}$-equivariant function $h$ and a scalar $\epsilon > 0$ there exists a set of integers $M_l \geq 1$, for $l \in [n]$, and $\mathcal{G}$-compatible functions $\{f^{l,i}\}_{i=1}^{M_l}$ such that*

$$\sup_{\boldsymbol{X} \in \mathbb{X}} \left| h_l(\boldsymbol{X}, \mathcal{G}) - \sum_{i=1}^{M_l} \phi\left(f^{l,i}(\boldsymbol{X}, \mathcal{G})\right) \right| < \epsilon, \tag{A10}$$

   *for all $l \in [n]$, where $h_l(\boldsymbol{X}, \mathcal{G}) \in \mathbb{R}$ denotes the lth component of $h$ and $\phi : \mathbb{R} \to \mathbb{R}$ is some function.*

*Proof:* The proof is based on a result presented in [8]. Let $\boldsymbol{W}$ denote a $n \times n$ adjacency matrix for graph $\mathcal{G}$ (*i.e.,* $w(u,v) = 0$ if the link $(u,v) \notin \mathcal{E}(\mathcal{G})$) and $w(u,v)$ denotes the $(u,v)$th element in $\boldsymbol{W}$. Let $\mathsf{W}$ denote the space of all such adjacency matrices $\boldsymbol{W}$ (for graph $\mathcal{G}$) such that $||\boldsymbol{W}||_\infty \leq 1$, *i.e.,* $|w(u,v)| \leq 1$ for all $u,v \in [n]$. Let $\mathsf{G}$ denote the set of all simple graph, *i.e.,* graphs with no self-loops or multi-edges.

For a $\boldsymbol{W} \in \mathsf{W}$ and a graph $\mathcal{H} \in \mathsf{G}$ define the function:

$$\mathrm{hom}\left(\mathcal{H}, \boldsymbol{W}\right) = \sum_{\pi \in \mathbb{M}(\mathcal{H}, \mathcal{G})} \prod_{u \in \mathcal{V}(\mathcal{H})} w(\pi(u), \pi(u)) \prod_{(u,v) \in \mathcal{E}(\mathcal{H})} w(\pi(u), \pi(v)). \tag{A11}$$

where $\mathbb{M}(\mathcal{H}, \mathcal{G})$ denotes the set of all maps $\pi$ from $\mathcal{V}(\mathcal{H})$ to $\mathcal{V}(\mathcal{G})$. Further, let $\mathcal{A}$ denote the following class of functions:

$$\mathcal{A} = \left\{ \boldsymbol{W} \to \sum_{\mathcal{H} \in \mathsf{H}} \alpha_{\mathcal{H}} \mathrm{hom}\left(\mathcal{H}, \boldsymbol{W} + 2\boldsymbol{I}\right) \; \middle| \; \begin{array}{c} \alpha_{\mathcal{H}} \in \mathbb{R} \\ \mathsf{H} \subset \mathsf{G} \\ \mathsf{H} \text{ is finite} \end{array} \right\},$$

where $\boldsymbol{I}$ denotes the $n \times n$ identity matrix. For a $\boldsymbol{W} \in \mathsf{W}$, graph $\mathcal{H} \in \mathsf{G}$, and a node $s \in \mathcal{V}(\mathcal{G})$ define the function $\mathrm{HOM}\left(\mathcal{H}, \boldsymbol{W}\right) \in \mathbb{R}^n$ such that its $s$th ($s \in \mathcal{V}(\mathcal{G})$) component is given by

$$\mathrm{HOM}_s\left(\mathcal{H}, \boldsymbol{W}\right) = \sum_{\substack{\pi \in \mathbb{M}(\mathcal{H}, \mathcal{G}), \\ \pi(1) = s}} \prod_{u \in \mathcal{V}(\mathcal{H})} w(\pi(u), \pi(u)) \prod_{(u,v) \in \mathcal{E}(\mathcal{H})} w(\pi(u), \pi(v)). \tag{A12}$$

Define the function space:

$$\bar{\mathcal{A}} = \left\{ \boldsymbol{W} \rightarrow \sum_{\mathcal{H} \in \mathsf{H}} \alpha_{\mathcal{H}} \mathrm{HOM}\left(\mathcal{H}, \boldsymbol{W} + 2\boldsymbol{I}\right) \;\middle|\; \begin{array}{l} \alpha_{\mathcal{H}} \in \mathbb{R} \\ \mathsf{H} \subset \mathsf{G} \\ \mathsf{H} \text{ is finite} \end{array} \right\}.$$

We have the following result from [8].

**Theorem 5 ([8])** *The following statements are true:*

1. *$\mathcal{A}$ is dense in the space of continuous $\mathcal{G}$-invariant functions.*

2. *$\bar{\mathcal{A}}$ is dense in the space of continuous $\mathcal{G}$-equivariant functions.*

*Proof:* The only difference between the spaces $\mathcal{A}$, $\bar{\mathcal{A}}$ in [8] and defined here is that here we fix the input graph $\mathcal{G}$ and restrict the space $\mathsf{W}$ to be the set of all weighted adjacency matrices of $\mathcal{G}$ (with bounded weights). However, the exact same arguments presented in [8] hold in this case towards establishing the statements in Theorem 5. $\square$

We now show how Theorem 5 can be translated to establish Theorem 4. We only present the arguments here for the $\mathcal{G}$-invariant case in Theorem 4, and the $\mathcal{G}$-equivariance case can be deduced using the same line of arguments.

Firstly, note that any hom $(\mathcal{H}, \boldsymbol{W})$ (in (A11)) can be written as:

$$\overline{\mathrm{hom}}\left(\mathcal{H}, \boldsymbol{X}\right) = \sum_{\pi \in \mathbb{I}(\mathcal{H}, \mathcal{G})} \prod_{u \in \mathcal{V}(\mathcal{H})} \theta_{\pi(u)}(\boldsymbol{x}_{\pi(u)}) \prod_{(u,v) \in \mathcal{E}(\mathcal{H})} \theta_{\pi(u), \pi(v)}(\boldsymbol{x}_{\pi(u)}, \boldsymbol{x}_{\pi(v)}), \qquad \text{(A13)}$$

for some input node features $\boldsymbol{X} = (\boldsymbol{x}_v)_{v \in \mathcal{V}(\mathcal{G})}$ and functions $\theta_u, \theta_{u,v}$ for all $u \in \mathcal{V}(\mathcal{G})$ and $(u,v) \in \mathcal{E}(\mathcal{G})$ such that $||\theta_u||_\infty \leq 1$ and $||\theta_{u,v}|| \leq 1$ (This follows from $||\boldsymbol{W}||_\infty \leq 1$). Furthermore, the reverse is also true, *i.e.,*, for every $\overline{\mathrm{hom}}\left(\mathcal{H}, \boldsymbol{X}\right)$ defined in (A13) there exists a weighted adjacency matrix $\boldsymbol{X}$, with $||\boldsymbol{W}||_\infty \leq 1$, such that $\overline{\mathrm{hom}}\left(\mathcal{H}, \boldsymbol{X}\right) = \mathrm{hom}\left(\mathcal{H}, \boldsymbol{W}\right)$ (define $w(u, u) = \theta_u(\boldsymbol{x}_u)$ and $w(u, v) = \theta_{u,v}(\boldsymbol{x}_u, \boldsymbol{x}_v)$ to get the required $\boldsymbol{W}$).

This observation, in conjunction with Theorem 5, shows that the set of functions

$$\mathcal{B} = \left\{ \boldsymbol{X} \rightarrow \sum_{\mathcal{H} \in \mathsf{H}} \alpha_{\mathcal{H}} \overline{\mathrm{hom}}\left(\mathcal{H}, \boldsymbol{X}\right) \;\middle|\; \begin{array}{c} \alpha_{\mathcal{H}} \in \mathbb{R}, \; \mathsf{H} \subset_{\mathrm{finite}} \mathsf{G} \\ \theta_u = \theta_u' + 2, \\ ||\theta_u||_\infty \leq 1, \text{ and} \\ ||\theta_{u,v}||_\infty \leq 1 \end{array} \right\},$$

is also dense in the space of continuous $\mathcal{G}$-invariant functions. We now show that every function in $\mathcal{B}$ can be written as a finite sum of $\mathcal{G}$-compatible functions composed with a non-linear function.

**Lemma 6** *For every $g \in \mathcal{B}$ there exists a finite set of $\mathcal{G}$-compatible functions $\{f^i\}_{i=1}^M$ and a non-linear function $\phi : \mathbb{R} \rightarrow \mathbb{R}$ such that*

$$g(\boldsymbol{X}) = \sum_{i=1}^{M} \phi\left(f^i(\boldsymbol{X})\right). \qquad \text{(A14)}$$

*Furthermore, $\phi$ are independent of $g \in \mathcal{B}$.*

*Proof:* A function $g \in \mathcal{B}$ is given by

$$g(\boldsymbol{X}) = \sum_{\mathcal{H} \in \mathsf{H}} \sum_{\pi \in \mathbb{M}(\mathcal{H}, \mathcal{G})} \alpha_{\mathcal{H}} \prod_{u \in \mathcal{V}(\mathcal{H})} \theta_{\pi(u)}(\boldsymbol{x}_{\pi(u)}) \prod_{(u,v) \in \mathcal{E}(\mathcal{H})} \theta_{\pi(u), \pi(v)}(\boldsymbol{x}_{\pi(u)}, \boldsymbol{x}_{\pi(v)}), \qquad \text{(A15)}$$

for some $\alpha_{\mathcal{H}}, \theta_u$, and $\theta_{u,v}$s. Note that the expression

$$\prod_{u \in \mathcal{V}(\mathcal{H})} \theta_{\pi(u)}(\boldsymbol{x}_{\pi(u)}) \prod_{(u,v) \in \mathcal{E}(\mathcal{H})} \theta_{\pi(u), \pi(v)}(\boldsymbol{x}_{\pi(u)}, \boldsymbol{x}_{\pi(v)}), \qquad \text{(A16)}$$

can be written as $\phi(f^{\mathcal{H},\pi}(\boldsymbol{X}))$ with $\phi(x) = \exp\{x\}$ and $f^{\mathcal{H},\pi}(\boldsymbol{X})$ a $\mathcal{G}$-compatible function given by

$$f^{\mathcal{H},\pi}(\boldsymbol{X}) = \sum_{u \in \mathcal{V}(\mathcal{G})} \log \theta_{\pi(u)}(\boldsymbol{x}_{\pi(u)}) + \sum_{(u,v) \in \mathcal{E}(\mathcal{G})} \log \theta_{\pi(u), \pi(v)}(\boldsymbol{x}_{\pi(u)}, \boldsymbol{x}_{\pi(v)}). \qquad \text{(A17)}$$

Thus, we have

$$g(\boldsymbol{X}) = \sum_{\mathcal{H} \in \mathsf{H}} \sum_{\pi \in \mathbb{M}(\mathcal{H}, \mathcal{G})} \phi\left(f^{\mathcal{H}, \pi}(\boldsymbol{X})\right), \tag{A18}$$

where we have modified $f^{\mathcal{H}, \pi}$ to incorporate the constant $\alpha_{\mathcal{H}}$. Since $\mathsf{H}$ and $\mathbb{M}(\mathcal{H}, \mathcal{G})$ are finite sets, we have the result. $\qquad\square$

The result in Theorem 4 follows from Lemma 6 and the observation that $\mathcal{B}$ is dense in the space of continuous $\mathcal{G}$-invariant functions.

$\qquad\square$

# D  Proof of Theorem 7

The proof is divided into four sub-sections. Here is a brief outline:

**1.** In Section D.1, we first prove an *aggregation lemma*. It (roughly) states the following: If the representation vectors at the root nodes of the H-tree $\mathcal{J}_\mathcal{G}$ are $\{h_r\}_{r \in R}$, at some iteration $t$, then in finitely many more message passing iterations it is possible to output a label $y_{v_0} = \sum_{r \in R} h_r$.

**2.** In Section D.2, we then prove that any $\mathcal{G}$-compatible function $f$ can be written as a sum $f(\boldsymbol{X}) = \sum_{r \in R} \gamma_r$ of component functions $\gamma_r$.

**3.** In Section D.3, we establish that the component functions $\gamma_r$ have a compositional structure that matches with the sub-tree $\mathcal{T}_r$ of the H-tree $\mathcal{J}_\mathcal{G}$ formed by the root node $r$ and its descendants. This helps in efficient computation of the component function $\gamma_r$ on the sub-tree $\mathcal{T}_r$.

**4.** The goal is to first estimate each component $\gamma_r$, by message passing on $\mathcal{T}_r$, and then aggregate by applying the aggregation lemma. In Section D.4, we put it all together to argue that it is indeed possible to approximate any (adequately smooth and bounded) compatibility function $f$, to arbitrary precision $\epsilon$, by the message passing described in (8). We obtain a bound on the number of parameters $N$ required to approximate any such function in Section D.4.

## D.1  Aggregation

Let the COMB function be a simple average function:

$$y_{v_0} = \text{COMB}\left(\{h_l^T \mid l \text{ leaf node in } \mathcal{J}_\mathcal{G} \text{ s.t. } \kappa(l) = v_0\}\right) \triangleq \frac{1}{|\{l \mid \kappa(l) = v_0\}|} \sum_{l : \kappa(l) = v_0} h_l^T, \quad \text{(A19)}$$

for some $T$, where index $l$ is over the set of leaf nodes in $\mathcal{J}_\mathcal{G}$. We first prove the following lemma.

**Lemma 7 (Aggregation)** *Let $h_r^t$ denote the representation vectors of root nodes $r \in R$ at some iteration $t$. If $h_r^t \in [0, 1]$ for all $r \in R$ and $\sum_{r \in R} h_r^t \in [0, 1]$, then there exists $t_0$ message passing iterations such that*

$$y_{v_0} = \sum_{r \in R} h_r^t, \quad \text{(A20)}$$

*for $T = t + t_0$. Further, the parameters used in this message passing and the number of iterations $t_0$ do not depend on $\{h_r^t\}_{r \in R}$.*

*Proof:* We first make a few assertions about the message passing described in (8), in the paper. The proof of the lemma directly follows from them. The assertions are self-evident and we only give a one line descriptive proof following its statement.

**Assertion 1.** Let $(v, u)$ be an edge in the H-tree $\mathcal{J}_\mathcal{G}$. If $h_v^{t-1} \in [0, 1]$ then there exists parameters $N_u$, $a_{u,t}^k$, $b_{u,t}^k$, and $\boldsymbol{w}_{u,t}^k$ in (8) such that

$$h_u^t = \text{AGG}_t\left(h_u^{t-1}, \{h_w^{t-1} \mid w \in \mathcal{N}_{\mathcal{J}_\mathcal{G}}(u)\}\right) = \text{ReLU}\left(\sum_{k=1}^{N_u} a_{u,t}^k \langle \boldsymbol{w}_{u,t}^k, h_{\bar{\mathcal{N}}(u)}^{t-1} \rangle + b_{u,t}^k\right),$$
$$= \text{ReLU}\left(h_v^{t-1}\right) = h_v^{t-1}. \quad \text{(A21)}$$

*The last equality holds only because $h_v^{t-1} \in [0, 1]$.*

**Assertion 2.** Let $(v, u)$ be an edge in $\mathcal{J}_\mathcal{G}$. If $h_u^{t-1} + h_v^{t-1} \in [0, 1]$ then there exists parameters $N_u$, $a_{u,t}^k$, $b_{u,t}^k$, and $\boldsymbol{w}_{u,t}^k$ in (8) such that

$$h_u^t = \text{AGG}_t\left(h_u^{t-1}, \{h_w^{t-1} \mid w \in \mathcal{N}_{\mathcal{J}_\mathcal{G}}(u)\}\right) = \text{ReLU}\left(\sum_{k=1}^{N_u} a_{u,t}^k \langle \boldsymbol{w}_{u,t}^k, h_{\bar{\mathcal{N}}(u)}^{t-1} \rangle + b_{u,t}^k\right),$$
$$= \text{ReLU}\left(h_u^{t-1} + h_v^{t-1}\right) = h_u^{t-1} + h_v^{t-1}. \quad \text{(A22)}$$

*The last equality holds only because $h_u^{t-1} + h_v^{t-1} \in [0, 1]$.*

**Assertion 3.** Let $\boldsymbol{h}_r^t$ denote representation vectors at root nodes $r \in R$ on the H-tree $\mathcal{J}_\mathcal{G}$ at some iteration $t$. If $\boldsymbol{h}_r^t \in [0,1]$ and $\sum_{r \in R} \boldsymbol{h}_r^t \in [0,1]$ then for any $r_0 \in R$ there exists $t_0$ message passing iterations, for some $t_0 > 0$, on the root nodes in $\mathcal{J}_\mathcal{G}$ such that $\boldsymbol{h}_{r_0}^{t+t_0} = \sum_{r \in R} \boldsymbol{h}_r^t$. Further, the parameters used in this message passing are independent of $\{\boldsymbol{h}_r^t\}_{r \in R}$.

*This can be established by looking at $\mathcal{T} = \mathcal{J}_\mathcal{G}[R]$ as a tree rooted at $r_0$ and performing message aggregation from the leaf nodes of $\mathcal{T}$ to the root node $r_0$ using Assertion 2.*

**Assertion 4.** If $\boldsymbol{h}_r^t \in [0,1]$ for some $r \in R$, then there exists $t_0$ message passing iterations from the root node $r$ to all the leaf nodes $l$ in H-tree $\mathcal{J}_\mathcal{G}$ such that $\boldsymbol{h}_l^{t+t_0} = \boldsymbol{h}_r^t$, for all leaf nodes $l$.

*This can be done by using Assertion 1 and successively passing the representation vector $\boldsymbol{h}_r^t$ from $r$ to all the leaf nodes $l$ in $\mathcal{J}_\mathcal{G}$.*

From Assertions 3 and 4 it is clear that, given $\{\boldsymbol{h}_r^t\}_{r \in R}$ at some $t$ (bounded in $[0,1]$ as described in the statement of the lemma), there exists $t_0$ message passing iterations such that $\boldsymbol{h}_l^{t+t_0} = \sum_{r \in R} \boldsymbol{h}_r^t$ at all the leaf nodes $l$ in $\mathcal{J}_\mathcal{G}$. Since the COMB operation computes a simple average (see (A19)) we have the result. $\square$

### D.2 Factorization

Next, we show that any compatibility function

$$f(\boldsymbol{X}) = \sum_{C \in \mathcal{C}(\mathcal{G})} \theta_C(\boldsymbol{x}_C), \tag{A23}$$

can be broken down into component functions $\{\gamma_r\}_{r \in R}$ such that

$$f(\boldsymbol{X}) = \sum_{r \in R} \gamma_r, \tag{A24}$$

where

$$\gamma_r = \sum_{C \in \mathcal{C}_r} \theta_C(\boldsymbol{x}_C), \tag{A25}$$

for all $r \in R$,[1] $\mathcal{C}_r$ are subsets of $\mathcal{C}(\mathcal{G})$ which form its partition, and $R$ is the set of root nodes in the H-tree $\mathcal{J}_\mathcal{G}$.

**Lemma 8 (Factorization)** *Let $f$ be a graph compatible function given in* (A23) *with its clique functions $\theta_C$. Then, for every $r \in R$ there exists a subset $\mathcal{C}_r \subset \mathcal{C}(\mathcal{G})$ such that*

$$\gamma_r = \sum_{C \in \mathcal{C}_r} \theta_C(x_C), \tag{A26}$$

*and $f(\boldsymbol{X}) = \sum_{r \in R} \gamma_r$. Further, the collection of subsets $\{\mathcal{C}_r\}_{r \in R}$ forms a partition of $\mathcal{C}(\mathcal{G})$, i.e., $\mathcal{C}_r \cap \mathcal{C}_{r'} = \emptyset$ whenever $r \neq r'$ and $\cup_{r \in R} \mathcal{C}_r = \mathcal{C}(\mathcal{G})$.*

*Proof:* Let $f$ be a graph compatible function given in (A23) with its clique functions $\theta_C$ and

$$(\mathcal{T}, \mathcal{B}) = \texttt{tree-decomposition}(\mathcal{G}), \tag{A27}$$

be the tree decomposition of graph $\mathcal{G}$. Note that the set of root nodes $R$, in the H-tree $\mathcal{J}_\mathcal{G}$, is in fact all the nodes in $\mathcal{T}$, namely $R = \mathcal{V}(\mathcal{T})$. Further, for every $r \in R$, $B_r \in \mathcal{B}$ is a bag of nodes $B_r \subset \mathcal{V}(\mathcal{G})$ associated with $r$.

It is known that for any clique $C$ in graph $\mathcal{G}$, *i.e.,* $C \in \mathcal{C}(\mathcal{G})$, there exists an $r \in R$ such that all nodes in $C$ are in the bag $B_r$, *i.e.,* $\mathcal{V}(C) \subset B_r$ [1]. However, it is possible that two bags $B_r$ and $B_{r'}$, for $r \neq r'$, may contain all the nodes of the same clique $C$.

Ideally, we would define

$$\mathcal{C}_r \triangleq \{C \in \mathcal{C}(\mathcal{G}) \mid \mathcal{V}(C) \subset B_r\}, \tag{A28}$$

---

[1]We omit the explicit dependence of the function $\gamma_r$ on $\boldsymbol{X}$ to ease the notation.

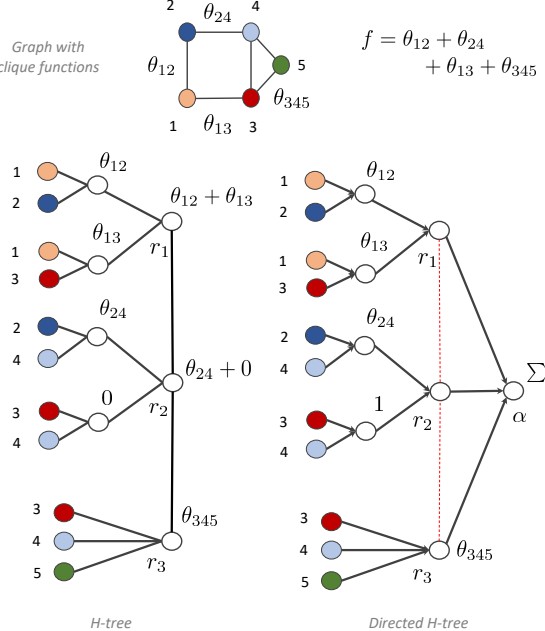

Figure A1: Shows the H-tree $\mathcal{J}_\mathcal{G}$ and the directed H-tree $\vec{\mathcal{J}}_\mathcal{G}$ of a graph. Computation of a compatible function $f$ is shown on the H-tree.

which is the set of all cliques $C$ in $\mathcal{G}$ such that all its nodes are in the bag $B_r$, and the functions $\gamma_r$ to be

$$\gamma_r = \sum_{C \in \mathcal{C}_r} \theta_C(\boldsymbol{x}_C), \tag{A29}$$

for all $r \in R$. However, this can lead the $\sum_{r \in R} \gamma_r$ to overestimate the function $f$. This is because two bags $B_r$ and $B_{r'}$ may contain all the nodes of the same clique $C$.

In order to avoid double counting of clique functions, we order the nodes in $R$ as $R = \{r_1, r_2, \ldots r_{|R|}\}$. We then iterate over these ordered $R$ nodes in the tree-decomposition to generate $\mathcal{C}_{r_k}$ and $\gamma_{r_k}$ (for $k = 1, 2, \ldots |R|$) as follows. Initialize $\mathcal{M}_1 = \emptyset$ and iterate over $k = 1, 2, \ldots |R|$:

$$\mathcal{C}_{r_k} = \{C \in \mathcal{C}(\mathcal{G}) \setminus \mathcal{M}_k \mid \mathcal{V}(C) \subset B_{r_k}\} \text{ and } \mathcal{M}_{k+1} = \mathcal{M}_k \cup \mathcal{C}_{r_k}, \tag{A30}$$

and set

$$\gamma_{r_k} = \sum_{C \in \mathcal{C}_{r_k}} \theta_C(\boldsymbol{x}_C), \tag{A31}$$

for $k = 1, 2, \ldots |R|$. This procedure ensures that we do not overestimate $f$ and have $f(\boldsymbol{X}) = \sum_{r \in R} \gamma_r$.

Furthermore, $\{\mathcal{C}_r\}_{r \in R}$ by its very construction (in (A30)) is pairwise disjoint and spans the entire $\mathcal{C}(\mathcal{G})$, thereby forming its partition. $\qquad \square$

### D.3 Compositional Structure

Fig. A1 illustrates computation of a compatible function on the H-tree. We see how the computation of $f$ splits as $f = \gamma_{r_1} + \gamma_{r_2} + \gamma_{r_3}$, where $\gamma_{r_1} = \theta_{12} + \theta_{13}$, $\gamma_{r_2} = \theta_{24}$, and $\gamma_{r_3} = \theta_{345}$. It is interesting to note that the functions $\gamma_r$, further, have a compositional structure that matches with the sub-tree induced by the root nodes $r$, and its descendants. For example, the compositional structure of $\gamma_{r_1}$ matches with the sub-tree formed by the root node $r_1$ and its descendants in the H-tree $\mathcal{J}_\mathcal{G}$. This turns out to be true in general for any compatible function $f$, and its factorization $\{\gamma_r\}_{r \in R}$ (in Lemma 8).

In order to make this precise, we introduce a few definitions that are inspired by [10, 9]. Let $\vec{\mathcal{D}}$ be a directed acyclic graph (DAG) with a single root node $\alpha$, i.e. all the directed paths in $\vec{\mathcal{D}}$ end at $\alpha$. We will use the term DAG to refer to a single-root DAG in this section.

**Definition 9** *A function $f : \boldsymbol{X} = (x_i)_{i \in [n]} \to f(\boldsymbol{X}) \in \mathbb{R}$ is said to have a compositional structure that matches with a DAG $\vec{\mathcal{D}}$, with root node $\alpha$, if the following holds:*

*1. Each leaf node $l$ of $\vec{\mathcal{D}}$ embeds one component of the input feature,* i.e., $\boldsymbol{h}_l = \boldsymbol{x}_i$ *for some $i \in [n]$.*

*2. For every non-leaf node $u$ there exists some function $\mathcal{H}_u$ such that*

$$\boldsymbol{h}_u = \mathcal{H}_u \left( \{ \boldsymbol{h}_w \mid w \in \mathcal{N}_{in}(u) \} \right), \tag{A32}$$

*where $\mathcal{N}_{in}(u)$ denotes the set of all incoming neighbors to node $u$.*

*3. $f(\boldsymbol{X}) = \boldsymbol{h}_\alpha$, where $\alpha$ is the single-root node of $\vec{\mathcal{D}}$.*

Let $\mathcal{T}_r$ denote the sub-tree of $\mathcal{J}_\mathcal{G}$ induced by the root $r \in R$ and its descendants in $\mathcal{J}_\mathcal{G}$. Further, let $\vec{\mathcal{T}}_r$ denote a directed version of $\mathcal{T}_r$ in which every edge $e$ in $\mathcal{T}_r$ is turned into a directed edge, pointing in the direction of the root node $r$. Note that $\vec{\mathcal{T}}_r$ is a DAG with node $r$ functioning as the single root node.

We now show that the components $\{\gamma_r\}_{r \in R}$ of the compatible function $f$ in Lemma 8 have a compositional structure that matches with $\vec{\mathcal{T}}_r$.

**Lemma 10 (Compositional Structure)** *The function $\gamma_r$, in Lemma 8, has a compositional structure that matches with the directed sub-tree $\vec{\mathcal{T}}_r$, for all $r \in R$.*

*Proof:* In Lemma 8, the function $\gamma_r$ is given by

$$\gamma_r = \sum_{C \in \mathcal{C}_r} \theta_C(\boldsymbol{x}_C), \tag{A33}$$

were $\mathcal{C}_r$ is given by

$$\mathcal{C}_r = \{ C \in \mathcal{C}(\mathcal{G}) \setminus \mathcal{M} \mid \mathcal{V}(C) \subset B_r \}, \tag{A34}$$

for some set $\mathcal{M} \subset \mathcal{C}(\mathcal{G})$. The set $\mathcal{C}_r$ can be thought of as a collection of cliques in the subgraph of $\mathcal{G}$ induced by the bag $B_r$, namely $\mathcal{G}[B_r]$. Therefore, the function $\gamma_r$ is a compatible function on $\mathcal{G}[B_r]$.

Note that, in Lemma 8, we showed that a graph $\mathcal{G}$ compatible function can be factored as a sum of $R$ functions, call them $\{\gamma_r\}_{r \in R}$, where $R$ is the set of nodes in the tree-decomposition $(\mathcal{T}, \mathcal{B})$. We have now argued that the functions $\gamma_r$ are compatible function on the subgraphs $\mathcal{G}[B_r]$.

Note that the set of all children of node $r$ in the sub-tree $\mathcal{T}_r$ (of the H-tree $\mathcal{J}_\mathcal{G}$) form a tree decomposition of $\mathcal{G}[B_r]$. This indicates that the function $\gamma_r$ should also split as a sum of functions, one corresponding to each node in the tree decomposition of $\mathcal{G}[B_r]$, by Lemma 8.

Thus, by successively applying Lemma 8, we can see that the compositional structure of $\gamma_r$ matches with the directed sub-tree $\vec{\mathcal{T}}_r$, constructed out of the H-tree $\mathcal{J}_\mathcal{G}$. $\qquad\square$

### D.4 Approximation

Lemmas 7 and 8 suggest that in order to approximate a compatible function $f(\boldsymbol{X}) = \sum_{C \in \mathcal{C}(\mathcal{G})} \theta_C(\boldsymbol{x}_C)$, with $f$ and $\theta_C$ bounded between $[0, 1]$, it suffices to generate representation vectors

$$\boldsymbol{h}_r^t \approx \gamma_r = \sum_{C \in \mathcal{C}_r} \theta_C(\boldsymbol{x}_C), \tag{A35}$$

at each root node $r \in R$ of the H-tree $\mathcal{J}_\mathcal{G}$, for some $t$. The approximation in (A35) must be such that

$$\left| \sum_{r \in R} \boldsymbol{h}_r^t - \sum_{r \in R} \gamma_r \right| = \left| \sum_{r \in R} \boldsymbol{h}_r^t - f(\boldsymbol{X}) \right| < \epsilon. \tag{A36}$$

Once such representation vectors $\boldsymbol{h}_r^t$ are generated at the root nodes of the H-tree, by Lemma 7, it's sum can be propagated to generate the node label $y_{v_0} = \sum_{r \in R} \boldsymbol{h}_r^t$, with message passing that is independent of the function being approximated.

Next, we show that the message passing defined in (8) in the main paper can indeed produce an approximation, give in (A36). The number of parameters required to attain this approximation will be an upper-bound on $N$.

To prove this, we consider a directed version of the H-tree $\mathcal{J}_{\mathcal{G}}$, where each edge in $\mathcal{J}_{\mathcal{G}}$ is turned into a directed edge pointing in the direction that leads to the root nodes $R \in \mathcal{J}_{\mathcal{G}}$. We also remove the edges between the root nodes $R$, and add another final node that aggregates information from all the root nodes. We call the final node the *aggregator* and call it $\alpha$. We call this directed graph $\vec{\mathcal{J}}_{\mathcal{G}}$. A directed H-tree graph is illustrated in Figure A1. The red colored edges between root nodes show the deleted edges between the root nodes $R$ in $\mathcal{J}_{\mathcal{G}}$ to get $\vec{\mathcal{J}}_{\mathcal{G}}$.

We assume that the messages propagate only in one direction, i.e. from the leaf nodes, where the input node features are embedded, to the aggregator node $\alpha$. We implement a shallow neural network at every non leaf node in $\vec{\mathcal{J}}_{\mathcal{G}}$, which takes in input from all its incoming edges, and propagates its output through its single outgoing edge, directed towards the root nodes.

This can be implemented in the original message passing (8) by setting the weight (*i.e.,* parameter $w_{u,t}^k$) component corresponding to the parent node, in the directed $\mathcal{J}_{\mathcal{G}}$, to zero. This final aggregation layer in $\vec{\mathcal{J}}_{\mathcal{G}}$ is only for mathematical purpose so that we can prove an $\epsilon$ approximation result, as in (A36).

With this, in the new message passing architecture on $\vec{\mathcal{J}}_{\mathcal{G}}$, each non-leaf node $u$ in $\vec{\mathcal{J}}_{\mathcal{G}}$ implements the following shallow neural network given by

$$\boldsymbol{h}_u = \mathrm{ReLU}\left(\sum_{k=1}^{N_u} a_u^k \langle \boldsymbol{w}_u^k, \boldsymbol{h}_{\mathcal{N}_{\mathrm{in}}(u)}\rangle + b_u^k\right), \tag{A37}$$

where $\boldsymbol{h}_{\mathcal{N}_{\mathrm{in}}(u)} \triangleq (\boldsymbol{h}_{u'} \mid u' \in \mathcal{N}_{\mathrm{in}}(u))$ denotes the vector formed by concatenating all the representation vectors $\boldsymbol{h}_{u'}$ of nodes $u'$ that have an incoming edge to $u$ in $\vec{\mathcal{J}}_{\mathcal{G}}$. Here, $a_u^k$ and $b_u^k$ are constants and $\boldsymbol{w}_u^k$ is a vector of size $d_u - 1$, which is the total number of incoming links to node $u$ in $\vec{\mathcal{J}}_{\mathcal{G}}$ and $d_u$ is the total number of links that node $u$ has in $\mathcal{J}_{\mathcal{G}}$. Thus, for every non-leaf node $u \in \vec{\mathcal{J}}_{\mathcal{G}}$ we have $(d_u - 1) \times N_u$ parameters that model the shallow network. The aggregator node generates the output by simply summing the representation vectors at the root nodes.

Note that, in (A37), $\boldsymbol{h}_u$ depends on the input node features $\boldsymbol{X}$. We omit this dependence in the notation for ease of presentation. We now define the space of functions that the above message passing on $\vec{\mathcal{J}}_{\mathcal{G}}$ produces:

$$\mathcal{F}(\mathcal{G}, N) = \left\{ \boldsymbol{X} \to \sum_{u \in R} \boldsymbol{h}_u \;\middle|\; \boldsymbol{h}_u \text{ given in (A37)} \right\}, \tag{A38}$$

where $N = \sum_u (d_u - 1) N_u$ is the sum of all the parameters used in (A37).

In the following, we will restrict ourselves to the DAG $\vec{\mathcal{J}}_{\mathcal{G}}$ and argue that any (smooth enough) function $f$ that has a compositional structure that matches with $\vec{\mathcal{J}}_{\mathcal{G}}$ can be approximated by a $g \in \mathcal{F}(\mathcal{G}, N)$ (see (A38)) with an arbitrary precision.

We now show that for any (smooth enough) function $f$, which has a compositional structure that matches with the directed H-tree $\vec{\mathcal{J}}_{\mathcal{G}}$, can be approximated by a $g \in \mathcal{F}(\mathcal{G}, N)$ (see (A38)) with an arbitrary precision.

**Theorem 11** *Let $f : [0,1]^n \to [0,1]$ be a function that has a compositional structure that matches with the DAG $\vec{\mathcal{J}}_{\mathcal{G}}$. Let every constituent function $\mathcal{H}_u$ of $f$ (see Definition 9) be $L_u$-Lipschitz with respect to the infinity norm. Then, for every $\epsilon > 0$ there exists a neural network $g \in \mathcal{F}(\mathcal{G}, N)$ such that $||f - g||_\infty < \epsilon$ and the number of parameters $N$ is bounded by*

$$N = \mathcal{O}\left( \sum_{u \in \mathcal{V}(\vec{\mathcal{J}}_{\mathcal{G}}) \setminus \{\alpha\}} (d_u - 1) \left(\frac{\epsilon}{L_u}\right)^{-(d_u - 1)} \right), \tag{A39}$$

*where $d_u$ denotes the degree (counting incoming and outgoing edges) for node $u$ in $\vec{\mathcal{J}}_{\mathcal{G}}$*

*Proof:* The proof of this result follows directly from the arguments presented for Theorem 3, Theorem 4, and Proposition 6 in [10, 9]. The first modification we make is the constant factor term $(d_u - 1)$ for each node $u$ in the summation in (A39). This appears here, but not in [10, 9], because in [10, 9] the node degree was considered as a constant. Here, the degree relates to the treewidth of the graph, and is an important parameter to track scalability of the architecture. The second modification is that we allow for different Lipschitz constants $L_u$ for different constituent function. However, the arguments in [10, 9] work for this case as well. □

We now apply Theorem 11 to the function $f$ given in the statement of Theorem 7. In it, $f$ is compatible with respect to $\mathcal{G}$. Thus, using Lemma 8 and Lemma 10, we can deduce that $f$ also has a compositional structure that matches with the directed H-tree $\vec{\mathcal{J}}_\mathcal{G}$. In Figure A1, we illustrate this for a simple example. Thus we can apply Theorem 11 on $f$ in order to seek an approximation $g \in \mathcal{F}(\mathcal{G}, N)$.

In applying Theorem 11, we see that the functions $\theta_C$ are 1-Lipschitz. Thus, all the nodes $u \in \vec{\mathcal{J}}_\mathcal{G}$ at which we compute $\theta_C$, $L_u = 1 \leq d_u - 1$. The remaining functions that are to be approximated on the $\vec{\mathcal{J}}_\mathcal{G}$ are the addition functions (see Figure A1 to know how they arise in computing a compatible function). In order to derive our result, it suffices to argue that a simple sum of $k$ variables, taking values in the unit cube $[0, 1]^k$, is $k$-Lipschitz with respect to the sup norm. This is indeed true and can be verified by simple arguments in analysis. Thus, for all the nodes $u$ on which we have to compute the addition, we have $L_u = d_u - 1$, where $d_u$ is the degree of node $u$ (counting both incoming and outgoing edges).

Putting all this together and applying Theorem 11 we obtain the result.

# E Proof of Corollary 8

We first obtain upper-bounds on the number of nodes $|\mathcal{V}(\mathcal{J}_\mathcal{G})|$ in the H-tree $\mathcal{J}_\mathcal{G}$ and the node degree $d_u$ for $u \in \mathcal{V}(\mathcal{J}_\mathcal{G})$. We prove the desired result by substituting these bounds in Theorem 7.

First, note that the subgraph of the H-tree $\mathcal{J}_\mathcal{G}$ induced by the set of root nodes $R$ is a tree decomposition $(\mathcal{T} = \mathcal{J}_\mathcal{G}[R], \mathcal{B})$ of $\mathcal{G}$, by construction; see Algorithm 1 (lines 1-3). Let tw $[\mathcal{J}_\mathcal{G}]$ denote the treewidth of the tree decomposition $(\mathcal{T} = \mathcal{J}_\mathcal{G}[R], \mathcal{B})$. Then the size of each bag $B_\tau \in \mathcal{B}$ is bounded by the treewidth tw $[\mathcal{J}_\mathcal{G}] + 1$ (see (5)). Let $T_\tau$ denote the sub-tree in $\mathcal{J}_\mathcal{G}$ that is formed by all the descendants of, and including, the node $\tau$ in $\mathcal{T} = \mathcal{J}_\mathcal{G}[R]$. Then, the number of nodes in $\mathcal{J}_\mathcal{G}$ is given by

$$|\mathcal{V}(\mathcal{J}_\mathcal{G})| = \sum_{\tau \in R} |\mathcal{V}(T_\tau)|. \tag{A40}$$

Note that the size of each sub-tree $|\mathcal{V}(T_\tau)|$ is bounded by

$$|\mathcal{V}(T_\tau)| \leq 1 + (\text{tw}\,[\mathcal{J}_\mathcal{G}] + 1)^{\text{tw}[\mathcal{J}_\mathcal{G}]+1}. \tag{A41}$$

This is because the depth of the tree $T_\tau$ is bounded by the bag size $|B_\tau|$, which is upper-bounded by tw $[\mathcal{J}_\mathcal{G}] + 1$. Further, no node in $T_\tau$ has a bag size larger than $|B_\tau|$ and therefore the number of children at each non-leaf node in $T_\tau$ is bounded by tw $[\mathcal{J}_\mathcal{G}] + 1$. The additional "+1" in (A41) accounts for the root node $\tau$ in $T_\tau$.

Finally, the number of nodes in the tree decomposition $\mathcal{T} = \mathcal{J}_\mathcal{G}[R]$ (or equivalently, the number of root nodes $R$) is upper-bounded by $n$, the total number of nodes in graph $\mathcal{G}$. This, along with (A40)-(A41), imply

$$|\mathcal{V}(\mathcal{J}_\mathcal{G})| \leq n + n\,(\text{tw}\,[\mathcal{J}_\mathcal{G}] + 1)^{\text{tw}[\mathcal{J}_\mathcal{G}]+1} \tag{A42}$$

Note that the degree minus 1, $d_u - 1$, is the size of the bag in a tree decomposition of some subgraph of $\mathcal{G}$. Since the size of the largest bag in the tree decomposition of the entire graph is bounded by tw $[\mathcal{J}_\mathcal{G}] + 1$, we have

$$d_u - 1 \leq \text{tw}\,[\mathcal{J}_\mathcal{G}] + 1, \tag{A43}$$

for all $u \in \mathcal{V}(\mathcal{J}_\mathcal{G})$.

Substituting (A43)-(A42) in (10) of Theorem 7 we obtain the result.

# F  Addendum to 3D Scene Graph Experiments

We provide more details on the (i) approaches and setup, (ii) the compute, train and test time requirements, (iii) the methods we use for tuning of our hyper-parameters, and (iv) the list of semantic labels in the dataset.

**Approaches and Setup.** We implement the neural tree architecture with four different aggregation functions $AGG_t$ specified in: GCN [5], GraphSAGE [7], GAT [12], GIN [13]. We randomly select 10% of the nodes for validation and 20% for testing. The hyper-parameters of the two approaches are separately tuned based on the best validation accuracy, while using all 70% of the remaining nodes for training. The READ function for the standard GNN (see (2)) is implemented as a single linear layer followed by a softmax. On the other hand, the COMB function (see (7)) for neural trees is implemented as a mean pooling operation, followed by a single linear layer and a softmax. We use different READ (resp. COMB) functions for the room nodes and the object nodes. We use the ReLU activation function and also implement dropout at each iteration. We train the architectures using the standard cross entropy loss function. The experiments are implemented using the PyTorch Geometric library.

**Time Requirements.** We study the time required for computing, training, and testing our model over the 3D scene graph dataset. It takes about $2.08$ sec to compute H-trees for all the 482 room-object scene graphs.

In Table A1, we report the train and test time for the standard GNN architectures – GCN, GraphSAGE, GAT, GIN – and the corresponding neural trees. We observe that the neural tree takes about 4x-10x more time to train compared to the corresponding standard GNN. This is expected because the H-tree is much larger than the input graph, and as a consequence, the neural tree architecture needs to train more weights than a standard GNN. The testing time for the neural

Table A1: Time Requirements: Train, and Test

| Model | Training (per epoch) | Testing |
|---|---|---|
| GCN | 0.072 s | 0.048 s |
| NT + GCN | 0.305 s | 0.058 s |
| GraphSAGE | 0.068 s | 0.042 s |
| NT + GraphSAGE | 0.311 s | 0.060 s |
| GAT | 0.089 s | 0.049 s |
| NT + GAT | 0.872 s | 0.107 s |
| GIN | 0.079 s | 0.043 s |
| NT + GIN | 0.348 s | 0.059 s |

trees, on the other hand, remains comparable to the standard GNN architectures. This makes the more accurate neural trees architecture amenable for real-time deployment. The reported times are measured when implementing the respective models on an Nvidia Quadro P4000 GPU processor.

**Hyper-parameter Tuning.** We tune the hyper-parameters in the following order, as recommended by [11]:

- Iterations: [1, 2, 3, 4, 5, 6]

- Hidden dimension: [16, 32, 64, 128, 256]

- Learning rate: [0.0005, 0.001, 0.005, 0.01]

- Dropout probability: [0.25, 0.5, 0.75]

- $L_2$ regularization strength: [0, 1e-4, 1e-3, 1e-2]

We first tune the number of iterations, hidden dimension, and learning rate using a grid search, while keeping dropout and $L_2$ regularization to the lowest value. For both standard GNN and neural tree, a single choice of the triplet: number of iterations, hidden dimension, and learning rate, yields significantly higher accuracy than the others. With this triplet fixed, we then tune the dropout and $L_2$ regularization using another grid search.

In training, we notice that the batch size does not have a noticeable impact on the training and test accuracy. After having experimented with various batch sizes between $32$ to $512$, we recommend and use a batch size of $128$ in our experiments.

Table 1 (in Section 7 of the main paper) reported the test accuracies for various standard GNNs and neural tree models. The tuned hyper-parameters for these models are given in Table A2. These hyper-parameters were tuned using the procedure described in the previous paragraph. A dropout

ratio of 0.25 turns out to be the optimal choice in all cases. The optimization is run for no more than 1000 epochs of SGD (using the Adam optimizer) to achieve reasonable convergence during training.

Apart from the four listed hyper-parameters (hidden dimension, number of iterations, $L_2$ regularization, learning rate), some of the implemented architectures (GAT, GraphSAGE, GIN) have their specific design choices and hyper-parameters. In the case of GAT, for example, we use 6

Table A2: Tuned Hyper-parameters for Various Models

| Model | hidden dim. | iter. | regularization | learning rate |
|---|---|---|---|---|
| GCN | 64 | 3 | 0.0 | 0.01 |
| NN + GCN | 128 | 4 | 0.0 | 0.01 |
| GraphSAGE | 128 | 3 | 1e-3 | 0.005 |
| NN + GraphSAGE | 128 | 4 | 1e-3 | 0.005 |
| GAT | 128 | 2 | 1e-4 | 0.001 |
| NN + GAT | 128 | 4 | 1e-4 | 0.0005 |
| GIN | 64 | 3 | 1e-3 | 0.005 |
| NN + GIN | 128 | 4 | 1e-3 | 0.005 |

attention heads and ELU activation function (instead of ReLU) to be consistent with the original paper. For GraphSAGE (in Table A2), we use the GraphSAGE-mean from the original paper, which does mean pooling after each convolution operation. In the case of GIN, we use the more general GIN-$\epsilon$ and train $\epsilon$ for better performance.

**Semantic Labels in the Dataset.** In the 482 room-object scene graphs we used for testing, the room labels are: bathroom, bedroom, corridor, dining_room, home_office, kitchen, living_room, storage_room, utility_room, lobby, playroom, staircase, closet, gym, garage. The object labels are: bottle, toilet, sink, plant, vase, chair, bed, tv, skateboard, couch, dining_table, handbag, keyboard, book, clock, microwave, oven, cup, bowl, refrigerator, cell_phone, laptop, bench, sports_ball, backpack, tie, suitcase, wine_glass, toaster, apple, knife, teddy_bear, remote, orange, bicycle.

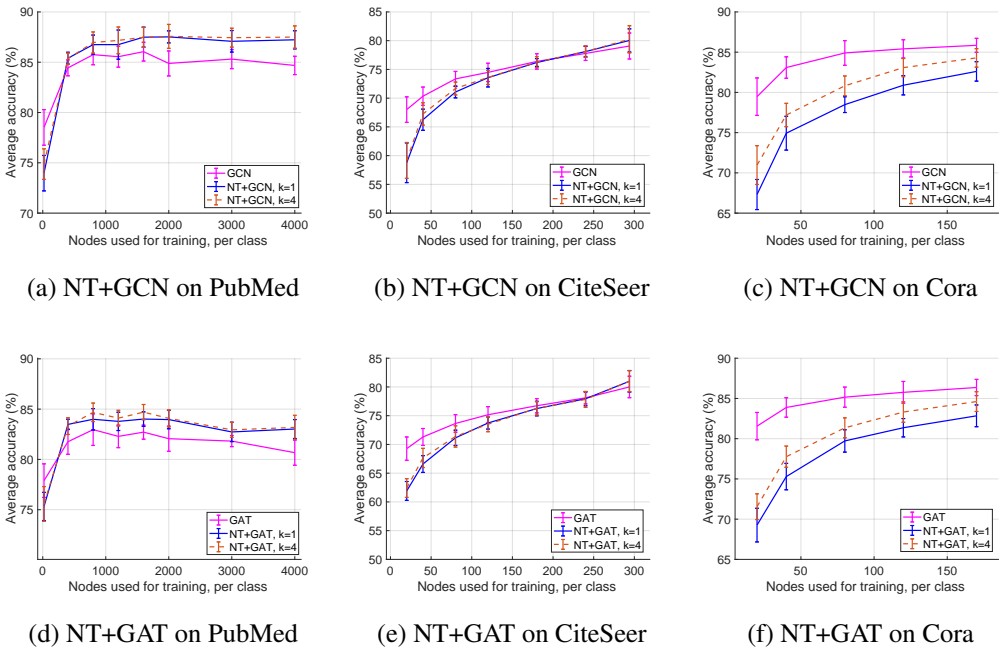

| (a) NT+GCN on PubMed | (b) NT+GCN on CiteSeer | (c) NT+GCN on Cora |
|---|---|---|
| (d) NT+GAT on PubMed | (e) NT+GAT on CiteSeer | (f) NT+GAT on Cora |

Figure A2: Accuracy vs training nodes (per label).

# G  Addendum to Citation Network Experiments

The goal of this section is two fold: (i) demonstrate the applicability of the neural tree architecture to large networks with high treewidth and (ii) show that the improved expressivity of the neural tree architecture becomes evident with increasing amount of training data.

**Datasets** We use three popular citation network datasets —PubMed, CiteSeer, and Cora [14]— where nodes are documents and undirected edges are citations. Each node has a class label representing the subject of the document. Table A3 outlines statistics about the dataset. The input citation network graphs have high treewidth, and therefore, are first sub-sampled (see Remark 6 in Section 5) using the bounded treewidth graph sub-sampling algorithm in [15], with a treewidth bound of $k$. The neural tree is then constructed from the sub-sampled graph.

Table A3: Citation network dataset statistics.

|  | PubMed | CiteSeer | Cora |
|---|---|---|---|
| Nodes | 19,717 | 3,327 | 2,708 |
| Edges | 44,338 | 4,732 | 5,429 |
| Classes | 3 | 6 | 7 |

**Approaches, Setup, and Hyper-parameters.** We implement the neural tree architecture with the aggregation function $AGG_t$ specified in: GCN [5] and GAT [12]. The READ function (see (2)) is implemented as a softmax, same as in [5, 12], and the COMB function (see (7)) is implemented as a mean pooling operation, followed by a softmax.

We use the same hyper-parameters (hidden dimension, number iterations, number of attention heads) for the neural trees as reported in the original GCN and GAT papers, except the learning rate, $L_2$ regularization, and dropout. These hyper-parameters pertain to the optimization algorithm used for training and are tuned to achieve the best results, *i.e.,* highest validation accuracy while not over-fitting. Better performance can be achieved using a specifically tailored message passing function for the neural trees, but the goal here is to understand when message passing on H-tree, *i.e.,* neural tree, performs better than message passing on the input graph, *i.e.,* standard GNN.

For each dataset, we randomly select 500 nodes for validation and 1000 nodes for testing. We vary the training data from 20 nodes per label, to all the remaining nodes (not used for validation and testing) in the network. We report the accuracy (and its variance) over 10 runs. The experiments are performed using PyTorch Geometric.

**Results.** Figure A2 plots test accuracy as a function of training data for all the three datasets. The test accuracy, for both standard GNNs and neural trees, increase with increasing number of training nodes. However, the increase tends to be much sharper for neural trees. Also note that, on the PubMed dataset, the test accuracy for the neural trees settles, after the sharp increase, to a value that is above the corresponding GNN architecture ((a) and (d) in Fig. A2). However, on the CiteSeer and Cora dataset, the test accuracy never really crosses the standard GNN architecture.

This is because the number of available training nodes (per label) is much less in the CiteSeer and Cora dataset, than it is in the PubMed dataset. In particular, CiteSeer and Cora has at most of 300 and 170 nodes for training (per label), respectively, while in PubMed we use at most 4000 nodes for training (per label) in our experiments. We see that neural trees perform better on PubMed, while the accuracy on CiteSeer is better than on Cora.

All this indicates that the performance of neural trees is directly proportional to the amount of available training data. While the standard GNNs can be expected to perform well when there is less available training data, the neural trees will most likely perform better in the high training data regime. We attribute this to the higher expressive power of the neural tree architecture. The neural tree architecture is able to seep in more data to yield higher prediction accuracy.

Another noticeable element in Figure A2 is the variation (or lack of it) in prediction accuracy in the treewidth bound $k$. Recall that the input graph is first sub-sampled using the bounded treewidth graph sub-sampling algorithm from [15] (see Remark 6 in Section 5). On the PubMed and CiteSeer dataset, we observe that the treewidth bound $k$ used for graph sub-sampling does not have much of an effect on the prediction accuracy. However,

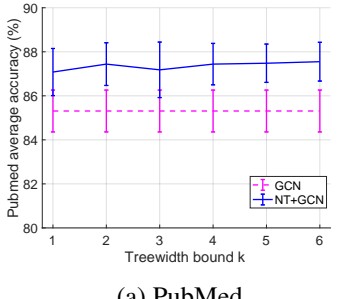 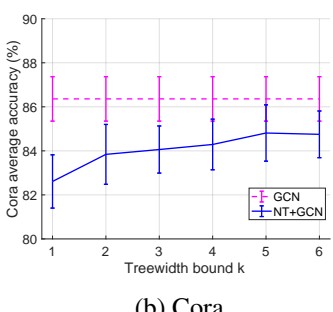

(a) PubMed           (b) Cora

Figure A3: Average accuracy as a function of treewidth bound $k$ for NT+GCN.

on Cora dataset, the performance can be improved by increasing the treewidth bound $k$. In Figure A3 we plot the average test accuracy as a function of treewidth bound $k$, for NT+GCN, on PubMed and Cora. In it, we use 3,000 and 170 nodes per label for training on the PubMed and Cora, respectively. While the prediction accuracy remains nearly the same on PubMed, there is a noticeable increase on Cora.

This indicates that in some datasets (*e.g.,* PubMed, CiteSeer) it is possible to retain the best possible performance, even after sub-sampling the input graph with a very low treewidth bound; say $k = 1$. This is very significant as it means that even if we disregard many of the existing edges in the network dataset, the performance does not degrade much. In the case of other datasets (*e.g.,* Cora), choosing a low treewidth bound $k$ serves as a good approximate solution. Note that the test accuracy gap between $k = 6$ and $k = 1$ is only about 2 percentage points, in Cora (see Figure A3).

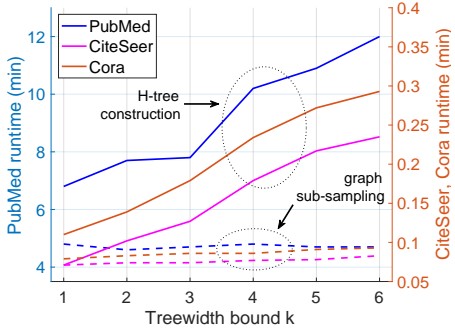

These results show that the neural tree architecture is a scalable architecture and can be applied to large networks with high treewidth. The choice of the treewidth bound $k$ will have to be tailored to the dataset in question. However, in order to achieve the full expressive power of neural trees, more training data is required.

Figure A4: Compute time (H-tree and graph sub-sampling) as a function of treewidth bound $k$.

**Time Requirements.** We study the time required to compute, train, and test our model over these large citation network datasets. The reported times are measured when implementing the respective models on an Nvidia Quadro P4000 GPU processor.

Figure A4 plots the time required (in minutes) for graph sub-sampling and H-tree construction. We see that while the time required for graph sub-sampling remains nearly the same, the time required for H-tree construction increases in the treewidth bound $k$. This is expected, as for larger $k$, the H-tree construction requires constructing tree-decompositions of many subgraphs of size at most $k$. The absolute numbers reported in Figure A4 can be improved as our current implementation uses the popular NetworkX library [2], which does not produce the time efficient implementation of many of the routines we use. However, we expect the trend observed in Figure A4 to hold true.

The increasing compute time with $k$ poses a trade-off between runtime and accuracy, especially for datasets like Cora, where increasing treewidth bound $k$ leads to increase in prediction accuracy.