# OpenReview forum: "Neural Trees for Learning on Graphs"
_NeurIPS.cc/2021/Conference — NeurIPS 2021 Poster_

### Official Review · Reviewer_pckR · 2021-07-04

**Rating:** 7
**Confidence:** 3

**Summary:**

The paper proposes a modification to GNN by first computing an H-tree and then running a GNN on it.

In the theoretical section they prove that this is a universal approximation for graph invariant functions which is a significant contribution, and show good empirical results.

**Limitations And Societal Impact:**

The authors clearly state the limitations of their method - graphs with  large tree width

**Main Review:**

I like the approach but my main concern is about the validity of the proof of theorem 7, I would gladly change my score if and when it is resolved.

- in equation 8 you define the aggregation but are not concerned with it being an invariant function as each edge has its own weight (also the common approach is to have w as a matrix not vector).

Because of this later in assertion 1 and 2 you assume you can send information from a specific edge, but that in itself contradicts invariance.
- If you claim invariance then this is a contradiction, if not then the theorem is not interesting as we could just use a MLP and approximate an invariant function using standard universality claims.

Other remarks:
- k-WL can be used efficiently for k=3, see "Provably Powerful Graph Networks".
- I found the 5.2 section very unclear and hard to follow.


**Time Spent Reviewing:**

7

---

> ### Author Response · Authors · 2021-08-10
> **Replying to all the major concerns raised**
>
> We would like to thank the reviewer for a careful reading of the manuscript and the valuable feedback.
>
> [On Theorem 7] Note that in Theorem 7, we do not approximate a G-invariant function but a G-compatible function. Therefore, all the assertions are valid.
>
> It is only in Theorem 2 that we argue that an invariant function can be approximated as a sum of finitely many G-compatible functions, composed with non-linear functions.
>
> Note that towards approximating a G-compatible function the H-tree does automatically selects node subsets and therefore enforces restrictions on weights W, as opposed to aggregating a feature vector for every subset of nodes in the graph and using a general MLP solution.
>
> [On MLP, Universality, and Neural Tree] The proposed architecture can approximate any G-compatible function with a parameter complexity that is exponential in the tree-width of the graph and linear in the number of nodes. We note that the complexity of the exact inference problem on a graphical model (a problem that is equivalent to approximating any G-compatible function) is also exponential in treewidth [13][46]. While it may be true that any MLP can approximate a G-compatible function, it does not do so with a bounded complexity that grows only exponential in treewidth, and hence, comparable to the minimum complexity of the problem. Neural tree architecture does just this.
>
> [Other remarks] We would like to thank the reviewer for pointing this out. We will change $k \geq  3$ to $k > 3$ in line 40 in the Introduction section. We will also re-work the writeup in 5.2 in the final manuscript if the paper is accepted.

---

> > ### Comment · Reviewer_pckR · 2021-08-14
> > **Question**
> >
> > Thanks for your feedback, seems like I misunderstood this section. After re-reading I understood what got me confused - In section 4 you talk about compatible graph function, in section 5 you jump to the seemingly unrelated neural tree model then finally show in section 6 that this can approximate compatible graph function and use that to thus is related to the expressivity of compatible graph function results from section 4.
> >
> > I think the flow should be made clearer and I don't think the introduction explain this well enough.
> >
> > Regardless, I am happy to raise my score once this misunderstanding was solved.

---

> > > ### Author Response · Authors · 2021-08-17
> > > **Reply**
> > >
> > > We would like to thank the reviewer for pointing out this lacuna in our presentation. We agree that the transition from Section 4 to Section 5 was abrupt and disconnected.
> > >
> > > In the final version, we will address this by adding a few lines, at the end of Sec 4 and towards the end of para 1 of Sec 5, explicitly stating the purpose of the proposed neural tree architecture (i.e. to efficiently approximate any G-compatible function). We will also hint at the result in Section 6 here so as to ensure that a reader does not think that Sec 5 (neural tree) is something unrelated to the results in Sec 4 and 6.
> > >
> > > In the Introduction, we will modify the contributions para 1-2 (lines 61-64), to ensure that there is further clarity on this and the flow is made clear to a reader.
> > >
> > > We hope that these changes will ensure no scope for confusion or misunderstanding in the aspect pointed out by the reviewer.
> > >
> > > We would again like to thank the reviewer for a thorough reading of the manuscript and for providing this feedback.

---

### Official Review · Reviewer_Xe3h · 2021-07-11

**Rating:** 7
**Confidence:** 4

**Summary:**

This work proposes a new GNN architecture based on performing message passing on the tree structures of the input graph, and theoretically and empirically demonstrates its strong expressive power.

**Ethics Review Area:**

["I don’t know"]

**Limitations And Societal Impact:**

Yes

**Main Review:**

Originality: This work proposes a novel GNN architecture which is very different from most existing GNN architectures. The key novelty of the new method is that the message passing is not performed on the input graph, but a hierarchical tree structure of the input graph. Such operations are demonstrated to be G-compatible, and equivalently be G-invariant/equivariant.

Quality: The authors have done empirical studies on 3d scene graph dataset and citation network dataset. The experiment results are very sound, showing the consistent performance boost over various of message passing mechanisms when using the proposed hierarchical tree structures. The ablation studies can technically support the theoretic claim of the new method.

Clarity: The paper is clearly written and well organized.

Significance: I personally think this is a very solid and enlightening work. It introduces the concept of G-compatible functions and theoretically demonstrate its relation with the expressive ability of graph neural networks, which is a unique and fundamental contribution in the related research area. I believe this work can attract more researchers to pay attention to G-compatible GNN structures in the future.

Considering the novelty, solid contribution, and good empirical studies, I recommend to accept this paper. My only concern is that the proposed method focuses on the semi-supervised node classification problem. I think the proposed method can be a general feature extractor for graph data. I am wondering can the proposed method be applied to graph-level problems (e.g., molecular graph classification)? If so, I believe additional experiments on such problems can make this work more comprehensive.


**Time Spent Reviewing:**

2

---

> ### Author Response · Authors · 2021-08-10
> **Review Response**
>
> We would like to thank the reviewer for a thorough reading of the manuscript and the positive feedback.
>
> [On feature extractor for graphs] Absolutely! The neural tree can be used to generate representation vectors for graphs and for the task of graph classification. We tried adding this discussion in the paper but that led to multiple threads and caused a lot of clutter in the presentation. We, therefore, left the discussion and results on graph representation, graph classification, and transductive generalization with neural trees out for a future paper, so as to simplify the presentation in this paper and convey the key idea behind the neural tree architecture.
>
> In light of the reviewer’s comments, we shall mention this in the concluding remarks, in the final version, if the paper is accepted.

---

> > ### Comment · Reviewer_Xe3h · 2021-08-17
> > **I am keeping my score**
> >
> > Thanks for the response. Overall, I think this work is solid and interesting. I am keeping my score and recommend an acceptance.

---

### Official Review · Reviewer_o8Qn · 2021-07-16

**Rating:** 8
**Confidence:** 3

**Summary:**


The work presented in this paper tackles the expressivity issue with graph neural networks (GNN). Drawing from the probabilistic graphical models' literature, the authors define the notion of G-compatible functions (i.e. can be factorized over the list of maximal cliques in graph G) and show how they are linked to G-invariant functions (i.e. invariant to node permutation). Then, the authors propose their new GNN architecture called Neural Tree. In their approach, an intermediate tree-structured representation, called H-tree, of the input graph G is constructed first. Traditional message passing is then performed on that H-tree instead of the input graph. The authors go on to show how neural trees can approximate G-compatible functions, thus capable of learning graph invariant functions.

Node classification experiments were conducted on two types of graphs: 3D scene graphs, and citation networks. The neural tree architecture was tested with four different aggregation functions (used during message passing) and compared against their typical architecture implementation (GCN, GraphSAGE, GAT, and GIN). Empirically, using the proposed neural tree architecture always yields higher accuracy. Additional experiments were done to measure the impact on the accuracy when changing the size of the training data (% of labeled nodes), and the number of iterations of message passing. Neural trees appear to be better when having access to a limited amount of labeled nodes, and the optimal number of iterations of message passing seems to follow a similar trend as with the standard GCN architecture.

**Limitations And Societal Impact:**

The authors have described a limitation of their method regarding large treewidth and suggested a graph subsampling technique to overcome that limitation. They show empirical evidence that the subsampling trick works on large graphs.

Regarding any potential negative societal impacts of their work, the authors said there is no direct impact. Since the approach heavily relies on tree-decomposition algorithms, in their experiments they used the junction-tree decomposition, I'm wondering if the authors are aware of any bias that such tree decomposition could have in the resulting tree (dropping rare edge, bias toward nodes with few/large neighbors, etc.)?

**Main Review:**

## Overview

To the extent of my knowledge, the proposed method is novel. The authors have tested it on standard nodes classification datasets against several techniques found in the literature. In addition, ablation studies were made to measure the impact of changing the size of the training data and the number of iterations of message passing. Moreover, the claims made in this paper are supported by both theoretical analysis and experimental results. To me, this submission appears to be technically sound. The paper is well written and organized, and it provides enough information for me to reproduce the results (Algorithm 1 + Supplementary material). In addition to advancing the state of the art, I believe this paper can encourage the community to design novel methods that are more mathematically sound. Overall, I recommend this paper for acceptance.


## What I like about this paper

- The proposed architecture is based on theory from probabilistic graphical models.
- The paper is clearly written and well-paced. The different concepts/definitions are introduced as you need them.


## Comments
- The paper focuses on undirected graphs, what about directed ones?
- At the end of section 7.1, it is mentioned that "optimal T is empirically close the average diameter of the H-Trees". Would it make sense to have an adaptive T, i.e. that varies according to the diameter of the H-tree provided in input?


-----
## Typos
- p.4, line 137: G-invaraint -> G-invariant

**Time Spent Reviewing:**

5

---

> ### Author Response · Authors · 2021-08-10
> **Review Response**
>
> We would like to thank the reviewer for a thorough reading of the manuscript and the positive feedback.
>
> [Directed Graphs] The proposed model can be used to approximate inference on the directed graph models as the joint distribution on a directed graphical model can also be described as a product of clique potentials. The method would be to first convert a directed model into an undirected model using the technique of moralization. The H-tree can then be constructed on this undirected graph, and not on the original directed graph.
>
> Thank you for this question. We will add this comment in the final version of the paper on this aspect.
>
> [Adaptive T] Yes, it absolutely would. Now that we know, from the ablation studies, that T is highly correlated to the diameter of the H-tree, we can in practice chose T to be some constant times the diameter, where the constant can be some number close to 1.
>
> In the experimental section, however, we chose a constant T and treated it as a hyper-parameter. The goal is to analyze T and bring out its structural relation with the H-tree.
>
> [Typo] Thanks for pointing this out. We will correct this in the final version of the paper.
>
> [Societal Impact] We would like to thank the reviewer for recommending this. Although intuitively it seems true that by dropping edges with graph-subsampling we may bias our prediction model, it is not possible for us to analyze its full impact in this work. We note that the proposed neural tree model does not rely on the specificities of the graph-subsampling procedure, except that it has smaller treewidth. One quick fix would be to generate multiple graph subsamples, by random initialization, and then aggregating the prediction at the end.
>
> However, this presents an opportunity and some open problems to the question of graph-subsampling: Is it possible to sub-sample an input graph to smaller treewidth so that the prediction model constructed from the subsampled graph, does not result in any bias?
>
> Incorporating the reviewer’s comments, we will add a section discussing the societal impact of the model and add the above comments in particular.

---

### Official Review · Reviewer_zPgP · 2021-07-22

**Rating:** 4
**Confidence:** 3

**Summary:**

This paper presents a new GNN architecture, which performs message passing on a hierarchical H-tree structure constructed from the original graph. Nodes in the H-tree represent subgraphs of the original graph, and thus message passing on H-tree can achieve higher expressive power by going beyond the node-to-node communication. Theoretical results show that the proposed neural tree architecture can approximate any smooth probability distribution function over an undirected graph. Such expressive power is achieved with number of parameters exponential with the width of the H-tree and linear with the original graph size. Experiments on a scene graph and a citation network show neural tree with existing GNN aggregation functions outperform the corresponding GNN alone.

**Limitations And Societal Impact:**

The paper does not discuss potential negative societal impact.

It may be worthwhile to discuss how the neural tree design can affect the energy consumption in training. As mentioned above, the design seems to come with much higher computation complexity than traditional GNNs.

**Main Review:**

## Originality

The idea of constructing an H-tree for GNN message passing seems quite original. The theoretical results on approximating any smooth probability distribution over undirected graphs also bring novel insights on the expressive power.

Although the idea of tree decomposition (e.g., constructing junction tree) and passing messages among tree nodes is known in the literature of probabilistic graphical models, it is valuable to extend such decomposition to the H-tree structure and further make connection to the GNN models.

## Quality

The overall algorithm design seems reasonable, although there does exist some part that seems unintuitive (see "other questions on the model design"). My main concerns on the design are with respective to scalability and generalization. My main concern on the experiments is whether they can scientifically validate the theoretical claims. Overall, even though the model enjoys some nice theoretical properties (under some strong / unrealistic assumptions, see "questions on the proof"), I doubt if it can have much practical use.

**Scalability and generalization**

From Theorem 7, the number of parameters of a neural tree is proportional to the number of nodes in the graph. From Eq 8, we also see that each node needs to have its own aggregation parameter $a_{u,t}$ and $b_{u,t}$. This makes the scaling of neural trees to large graphs quite challenging. Intuitively, this can also hurt generalization, since the learned model is so tightly coupled with the specific input graph structure. The experiments also seem to indicate the issue in generalization (small number of training nodes corresponds to lower accuracy of NT-GCN than GCN). Note that this is not the case for other GNN models. For example, GIN can establish the equivalence with 1-WL by sharing all weight parameters among all nodes. Overall, I feel that letting each node have its own parameter seems to be a restrictive setup, and this may make the theoretical results less interesting.

As for the scalability, there are many factors in the design that may hurt scalability to large graphs. For example, the total number of parameters need to be exponential with the tree width and linear with the graph size. It is argued that the tree width can be practically reduced by subsampling. However, the expressivity analysis does not hold under subsampling, and the tree width may grow with the graph size. Even the tree decomposition step can be quite expensive: the $O(E(k^2+ V))$ complexity is polynomial with the graph size, but can be practically very expensive already (e.g., consider the medium scale graphs with 1 million nodes in the Open Graph Benchmark).

In addition, it would be nice if the authors can profile the total training time to understand the tradeoff between computation complexity and accuracy for the proposed NT model.

**Questions on the proof**

For the supporting Lemma 6 of Theorem 7, the assumption that $\Sigma h$ is bounded within 0 and 1 seems quite unrealistic. Each $h$ is between 0 and 1, and how can the sum be bounded? Is it even a valid assumption?

For Theorem 3, how can we directly conclude from Eq A17 that $f$ is $G$-compatible? In other words, there seems to be a missing connection with max cliques of the graph?

**Questions on the experiments**

Please clarify the aggregation function of NT-GCN used in experiments. Does the aggregation follow Equation 8, or does it have parameter sharing among nodes? If NT-GCN does not have different $a_{u,t}$ and $b_{u,t}$ for each $u$, then I regard this as major discrepancy between the theoretical and experimental setups. The experiments thus cannot verify the analysis on expressive power, and also can hardly resolve the concern on generalization.

The scalability and generalization concerns are similarly posed to the experiments. The evaluated datasets are both very small, indicating the challenge in scalability.

As for baselines, more expressive models such as GIN and GAT may be more reasonable than GCN, since the main claim in the paper is w.r.t. improvement on expressive power. Therefore, for the citation network, it may be better to also include evaluation on GIN, GraphSAGE and GAT.

**Other questions on the model design**

I wonder the intuition behind applying aggregators such as GCN on H-tree. On the original graph, the GCN can learn its spectral property, and the aggregation operation can be understood as smoothing of the node features. However, with NT-GCN, what are the meaningful spectral properties associated with the H-tree structure? In addition, since the H-tree nodes follow an explicit hierarchical relation (e.g., leaf and root), would it be more reasonable that the aggregation makes use of such hierarchy? In the current design, the aggregation doesn't differentiate if a neighbor is a root or a leaf.

What is the relation between the design by Eq 8 and the other designs such as NT-GCN and NT-GraphSAGE? For example, for NT-GraphSAGE, does it still have a learnable parameter, $a_{u,t}$, distinct for every node $u$, or does it simply perform mean of features.


## Clarity

The paper is clearly written. The neural tree design is illustrated with examples, which helps with understanding. It would be better to clarify a few doubts on the theoretical results, as listed above.


## Significance

On the positive side, the design connects probabilistic graphical models with GNNs, and can possibly lead to future exploration.

The biggest factors that limit the significance of neural tree are the scalability and generalization. Even though the design may enjoy some nice theoretical properties, I am not convinced that it can significantly advance the current models that are used in practice.

In its current form, I'm not sure if the theorems are correct. The experiments also doesn't seem to validate the theorems due to quite different setup. If the authors can clarify the two questions in "Questions on the proof", and the NT-GCN question in "Questions on the experiments", I am willing to increase the score.

**Time Spent Reviewing:**

10

---

> ### Author Response · Authors · 2021-08-10
> **Replying to all the major concerns raised**
>
> We would like to thank the reviewer for a thorough reading of the manuscript and for providing valuable comments. We are replying to all the concerns raised by the reviewer.
>
> [Originality] Thanks for the positive comments about the originality of the idea, the relevance of the connections we establish with probabilistic graphical models, and the novelty of our theoretical results.
>
> [Quality and Clarify] Thanks for appreciating the design and the clarity of the manuscript – we address the reviewer’s comments about scalability and the theoretical results below.
>
> [On Complexity Comment] We would like to note here that the underlying complexity of many graph problems – such as exact inference on graphical models, the graph isomorphism problem (and equivalently approximating G-invariant function) – is known to be exponential in the treewidth of the graph and polynomial in the number of nodes [13][46][R1]. Therefore, no graph neural network architecture that relies on local message passing – how so ever simple and scalable they may be – can approximate solutions to these hard problems. If they could, then this would have to overturn the well-known complexity results of the underlying problems (problems such as exact inference and graph isomorphism).
>
> In the neural tree, we propose an architecture that can provide a solution to these problems (exact inference in particular) in parameter complexity that is exponential in treewidth – a complexity of proportionate order to the actual problem complexity. This can be seen validated in the experimental section, for instance in Figure 3, where the average accuracy for a traditional GNN caps out, while that for the corresponding neural tree keeps increasing.
>
> [Generalizability Comment] We do not think as the comments that the neural tree necessarily results in worse generalizability. On the contrary, when 70% of node labels are known at training, in Figure 3, the neural tree yields better prediction accuracy over the traditional GNN architecture on unseen data. Does this not show the generalizability of the proposed architecture?
>
> The differentiator between the neural tree and the traditional GNN architectures is not generalizability, but that the former can provide for better approximations to the underlying problems (such a graph inference) with increasing training data.
>
> It is true that this comes at the cost of computation and scalability, but this limitation is inherent to the underlying problem, i.e. exact inference on a graph or approximating G-compatible function has a computational complexity that grows exponentially in the treewidth of the graph. It is not possible to approximate any G-compatible function with a local message passing GNN. The proposed architecture provides for a way to do this with a parameter complexity that is of the same order as the complexity of the underlying problem.
>
> [On Parameter Sharing Comment] We would like to mention that, contrary to the reviewer's reading, we do not assume that each node has its own aggregation parameter in the neural tree architecture. The correct equations for the message propagation in the neural tree are given in equations (6)-(7).  Equation (8) is presented for mathematical convenience and can be implemented in (6) by choosing an appropriately higher hidden dimension. We note this in footnote 1. In the experimental section, we have parameter sharing (for the message passing function) across all nodes in the H-tree. In the final version, we plan to elaborate the footnote 1 as this seems to be the source of confusion.
>
> All the experimental results use the same parameters for message passing and are in conformity with equations (6)-(7). We do agree with the reviewer that the performance can be improved by choosing more tailored message passing functions, in which the parent and children nodes are treated separately in the message passing (6)-(7). However, the goal of our experimental results was to demonstrate that message passing on the H-tree, using the same type of message passing function as used on the graph in GNN, leads to noticeable improvements in performance.
>
> We hope that this convinces the reviewer regarding the question raised on the experiments.
>
> [On Lemma 6 Assumption] In the statement of the Lemma, we assume $h_{r}$ and $\sum_{r \in R} h_{r}$ to be both bounded between 0 and 1. We do not assert that one is implied by the other. We would like to make a note here that functions the bounded between $[0, 1]$ are for mathematical convenience only, and that the result can be extended to any functions that are bounded between $[0, B]$, for any $B > 0$. In the final version, we will add comments to make sure that this confusion is averted.
>
> [On A17] In A17, $f$ is expressed only as a function of individual node features $x_{\pi(u)}$ and pairwise node features $(x_{\pi(u)}, x_{\pi(v)})$. Furthermore, the pairwise function $\theta_{\pi(u), \pi(v)} = 0$ if $(\pi(u), \pi(v))$ is not an edge in graph $\mathcal{G}$. Therefore, f in A17 is a G-compatible function. It only uses cliques of size 2 (i.e. edges in G) to approximate $f$.
>
> We will add a sentence after A17-18 to explain this more clearly. We hope that this convinces the reviewer that the proof of the theorems is correct.
>
> [Training time, societal impact] We do provide the training time requirements in the supplementary material (see Table A1) and computation time requirements to compute H-tree, as well as perform graph sub-sampling (see Figure A4). To summarize we observe that it takes about 4x-10x more time to train a neural tree compared to the corresponding GNN. However, this cost yields increased performance in prediction accuracy. Therefore, there is a tradeoff between compute time, training time, energy consumption on one hand, and the prediction accuracy on the other. In applications such as citation networks, one may favor saving energy over prediction accuracy, while in applications such as robotics and 3D scene graphs, loss in prediction accuracy can have disastrous effects (ex. unable to detect and correctly classify an object in front of a self-driving car) and one may favor expending the extra bit of energy for it.
>
> We would like to thank the reviewer for this comment. We will add a section on societal impact and discuss this tradeoff and its implications in more detail.
>
> [R1] Martin Grohe, Daniel Neuen, Pascal Schweitzer, and Daniel Wiebking. 2020. An Improved Isomorphism Test for Bounded-tree-width Graphs. ACM Trans. Algorithms 16, 3, Article 34 (June 2020), 31 pages. DOI:https://doi.org/10.1145/3382082

---

> > ### Comment · Reviewer_zPgP · 2021-08-20
> > **Still Not Convinced on the Theoretical Part**
> >
> > First of all, thanks for the detailed response. However, after reading your explanation and the paper carefully again, I am still not convinced about some theoretical results.
> >
> > ## On parameter sharing
> >
> > You mentioned that Eq 8 can be implemented by Eq 6 by a higher dimension. Sure. But the key question is how high the dimension needs to be. For example, if we allow the dimension to be of O(E), then essentially it is just equivalent to have a dedicated parameter for each edge. In other words, if we need the dimension to grow / be dependent on graph size, it is not a true / meaningful parameter sharing scheme.
> >
> > Then I also wonder how useful the model is if it requires such large number of parameters. I agree that it may be theoretically needed to approximate any G-compatible functions. However, I can hardly imagine why it can benefit the practical tasks such as node classification (e.g., those evaluated in experiments). Normally, a good node classifier interpolates unseen labels by learning high level features such as homophily or salient subgraph structures. The former can be learned from the smoothing operation by GCN-type models and the later can be solved by simple message passings of GIN-type models. Neither requires to scale the model size with the graph size.
> >
> > So I feel there is a gap between theoretical results and the experiments. The relatively simple task of node classification doesn't seem to justify the need of such a complicated and expensive neural tree model. At least, if you want to argue node classification is a hard task and the expressiveness of neural tree is critical, you need to evaluate more expressive baselines such as GAT (which in some sense is similar to, but less powerful than, neural tree as GAT allows weights on each edge).
> >
> > ## On Lemma 6
> >
> > I probably should rephrase my original question, since my real concern is how Lemma 6 can lead to the main theorem of the paper. I agree that $\sum h$ being bounded is an assumption to Lemma 6. So I agree Lemma 6 itself is correct. But with such an assumption, it means Lemma 6 does not hold for any input $h$ in $[0, 1]$. On the contrary, the main theorems (Theorem 2 and 7) is based on all features in $[0, 1]$. I wonder how such assumption of Lemma 6 is addressed in proving the general statement of Theorem 2 and 7.
> >
> > I may be missing some details in the proof. So I would love to follow up on this issue.
> >
> > I will follow up on the other parts of your response once the above two points are clarified.
> >
> > Thanks!

---

> > > ### Author Response · Authors · 2021-08-25
> > > **Reply**
> > >
> > > # On parameter sharing
> > >
> > > ### $\textbf{Comment:}$You mentioned that Eq 8 can be implemented by Eq 6 by a higher dimension. Sure. But the key question is how high the dimension needs to be. For example, if we allow the dimension to be of O(E), then essentially it is just equivalent to have a dedicated parameter for each edge. In other words, if we need the dimension to grow / be dependent on graph size, it is not a true / meaningful parameter sharing scheme.
> > >
> > > $\textbf{Reply:}$ In Theorem 7 and Corollary 8, we bound the total number of parameters that are needed and not the hidden dimension. We believe this is the most important model complexity metric that one needs to be concerned of when designing a new architecture.
> > >
> > > In applying the neural tree architecture to node prediction on 3D scene graphs (see Sec 7.1), we are able to outperform local message passing GNN architecture significantly with a reasonable choice of hidden dimension. The hyperparameter tuning searches over hidden dimensions h_{dim} of [16, 32, 64, 128, 256]. We observe that for h_{dim} >= 32 there isn’t any significant improvement in performance, i.e. we are able to attain the best prediction accuracy (there is a small improvement – and therefore, we report the best as 64 and 128).
> > >
> > > This suggests that the proposed architecture is not impractical and we can find hidden dimension in a practical setting so that neural tree outperforms traditional GNNs. If the reviewer suggests, we can include this ablation study – which plots our prediction model accuracy w.r.t the hidden dimension.
> > >
> > > However, the question on obtaining a bound on the hidden dimension as a function of network size is an interesting one. Unfortunately, we do not address it in this paper but it does pique our curiosity. We will investigate this in the future as an extension of this work.
> > >
> > > ### $\textbf{Comment:}$ Then I also wonder how useful the model is if it requires such large number of parameters. I agree that it may be theoretically needed to approximate any G-compatible functions. However, I can hardly imagine why it can benefit the practical tasks such as node classification (e.g., those evaluated in experiments).
> > >
> > > $\textbf{Reply:}$ We observe that our experiments on 3D scene graphs (see Sec 7.1 and Table 1) show otherwise. Message passing on neural trees increases the prediction accuracy on the 3D scene graphs. The choice of hidden dimensions is reported in the supplementary material (see Sec. F). This suggests that, in practice, it is possible to get a reasonable choice of hidden dimension and outperform the traditional GNN architectures.
> > >
> > > ### $\textbf{Comment:}$ Normally, a good node classifier interpolates unseen labels by learning high level features such as homophily or salient subgraph structures. The former can be learned from the smoothing operation by GCN-type models and the later can be solved by simple message passing of GIN-type models. Neither requires to scale the model size with the graph size.
> > >
> > > $\textbf{Reply:}$ Yes, but such “good” architectures have been shown to be unable to solve the graph isomorphism problem (GIN for instance), nor can they approximate G-compatible function, i.e., they cannot approximate inference on the graph. Would the existing architectures still be “good” if they perform poorly – in theory and in practice? Table 1 shows that the prediction accuracy with message passing on neural trees is higher than doing message passing on graphs.
> > >
> > > The traditional message passing architectures on graph are low complexity – no doubt! But, they do not and cannot approximate graph compatible function, solve graph isomorphism, or graph equivariant functions. Note that the complexity of approximating inference on graphs, and hence a graph compatible function, is exponential in treewidth. We have an architecture that does this with the same parameter complexity. Applying neural trees to node classification task in Sec 7.1 shows its practicability.
> > >
> > > ### $\textbf{Comment:}$ So, I feel there is a gap between theoretical results and the experiments. The relatively simple task of node classification doesn't seem to justify the need of such a complicated and expensive neural tree model.
> > >
> > > $\textbf{Reply:}$ Our experimental results with 3D scene graphs (see Sec. 7.1) indicate otherwise. The improvement obtained when using the proposed Neural Tree can be substantial (see Table 1).
> > >
> > > ### $\textbf{Comment:}$ At least, if you want to argue node classification is a hard task and the expressiveness of neural tree is critical, you need to evaluate more expressive baselines such as GAT (which in some sense is similar to, but less powerful than, neural tree as GAT allows weights on each edge).
> > >
> > > $\textbf{Reply:}$ We remark that the paper already evaluates the GAT baseline (see Sec. 7.1). We observe that neural tree outperforms GAT on the input graph by a good 16 percentage points (see Table 1).
> > >
> > > # On Lemma 6
> > >
> > > ### $\textbf{Comment:}$ I probably should rephrase my original question, since my real concern is how Lemma 6 can lead to the main theorem of the paper. I agree that $\sum h$ being bounded is an assumption to Lemma 6. So, I agree Lemma 6 itself is correct.
> > >
> > > $\textbf{Reply:}$ Thank you! We are glad that we could clarify this misunderstanding.
> > >
> > > ### $\textbf{Comment:}$ But with such an assumption, it means Lemma 6 does not hold for any input h in [0,1]. On the contrary, the main theorems (Theorem 2 and 7) is based on all features in [0,1]. I wonder how such assumption of Lemma 6 is addressed in proving the general statement of Theorem 2 and 7.
> > >
> > > $\textbf{Reply:}$ Thank you for this comment. Please note the second line in the statement of Theorem 7, which states that the clique functions $\theta_c$ in Definition 1 are also bounded between [0, 1].
> > >
> > > The proof of Theorem 2 is in Appendix C and does not use Lemma 6.
> > >
> > > ### $\textbf{Comment:}$ I may be missing some details in the proof. So, I would love to follow up on this issue.
> > >
> > > $\textbf{Reply:}$ We would like to thank the reviewer for a thorough reading of the manuscript and their valuable comments.

---

> > > > ### Comment · Reviewer_zPgP · 2021-09-01
> > > > **Final Thoughts**
> > > >
> > > > I thank the authors for the detailed responses and follow-ups.
> > > >
> > > > Reading the paper carefully the third time, I would like to summarize my final conclusion.
> > > >
> > > > ## My misunderstandings addressed
> > > >
> > > > My comment on the $[0,1]$ bound of Lemma 6 and Theorem 7 may not be valid. Therefore, I am no longer concerned on the correctness of Theorem 7. I thank the authors for pointing out the assumptions on $\theta_c$ stated in Theorem 7.
> > > >
> > > > ## My concerns that still persist
> > > >
> > > > ### Parameter sharing
> > > >
> > > > Unfortunately, I still don't think Neural Tree implements a true parameter sharing scheme. As I mentioned in my previous follow-up, even if NT propagation can be written in the form of the normal GNN propagation, the hidden dimension may still need to grow with the graph size -- This somehow contradicts with the normal GNN design philosophy and would have significant impact on scalability (not as the authors claimed).
> > > >
> > > > The authors' response to my follow-up actually verifies my understanding. I agree that the total number of parameters is equally interesting as the hidden dimension size. However, as Theorems 7 and 8 suggest, the total number of parameters of NT grows with the graph size (or, equivalently, the tree size). On reading the proof in the Appendix, I find that "each edge having a dedicated weigh parameter" is critical in derivation many steps. e.g., in propagating the directed H-tree, you need to obey the parent-child hierarchy and explicitly set the parent-child weight to 0 (line 248, Appendix). However, in other steps, the propagation needs to ignore such hierarchy to re-enable the parent-child propagation. I don't think there is any parameter sharing scheme that can implement such propagation pattern.
> > > >
> > > > ### Discrepancy between theory and experiments
> > > >
> > > > The authors mention that in experiments, they do not use dedicated weight parameters for each edge, and they implement a parameter sharing scheme with small hidden dimension.
> > > >
> > > > It is hard to say whether such "small hidden dimension" is good or bad. From practical perspective, surely, a small hidden dimension indicates feasibility in real-life applications. On the other hand, in a scientific view, we shouldn't use such a hidden dimension setting. There is no theoretical understanding on what NT can express with such small hidden dimension (as suggested by the proof in Appendix). So it becomes unclear where the empirical accuracy improvement comes from.
> > > >
> > > > Also, as mentioned in my initial review, I am not fully convinced on the generalization ability of NT. Intuitively, such a large parameter size can easily lead to overfitting. I agree with the authors that in experiments, there is little evidence of poor generalization. But could this just be because of the "small dimension shared parameter" setup in the experiments?
> > > >
> > > > ### Scalability
> > > >
> > > > Scalability is one of the main claims. However, I am not convinced that NT is scalable. Depending on the context, "scalability" can have very different criteria. Comparing with "exact graphical inference" or "exact isomorphism test" (as replied by authors), I agree that NT can be scalable. Yet in the context of this paper, I think a more appropriate comparison should be made with the normal GNNs -- then the complexity (in terms of both parameters and computation) of NT is way too high, and I don't think it is true that NT is scalable.
> > > >
> > > > ### Relation with GNN
> > > >
> > > > This is related with the point on parameter sharing. I feel that the connection between NT and GNNs is not very well established. Without parameter sharing, the only commonality between NT and GNN is the iterative message passing operation. However, parameter sharing is actually a very important characteristic of GNNs, since message passing alone is seen in many non-GNN algorithms as well.
> > > >
> > > > ### Significance of experimental results
> > > >
> > > > As mentioned in my initial review and follow-up, I would like to see comparison with more expressive baselines such as GAT. I thank the authors for pointing out existing GAT results on the 3D scene graph.
> > > >
> > > > However, it would be much more convincing if results of those baselines can be shown on the citation network. The 3D scene graph is way too small. The graphs for different rooms are not connected, meaning that the average graph size is only $1 + 2338 / 482 < 6$.
> > > >
> > > > Recall the reason for my insisting on the GAT results: I wonder if the potential accuracy benefit of NT can justify the dramatically increased parameter and computation complexity -- especially when the less powerful but much more efficient GNNs can already achieve very good accuracy on many node classification tasks. The tiny 3D scene graphs won't exhibit any complexity issue anyways. So even though the accuracy gain is significant, these results do not fully address my scalability concerns.
> > > >
> > > > ## Summary
> > > >
> > > > I have mixed feelings on this paper. As mentioned in my initial review, I like the originality of the neural tree design. However, I do have many concerns raised from the non-parameter sharing design. For pure theoretical analysis, it is fine to assume each edge has a dedicated weight parameter. However, when you compare with the normal GNNs, this does seem to me a fundamental limitation.

---

> > > > > ### Author Response · Authors · 2021-09-01
> > > > > **Thanks and Summary**
> > > > >
> > > > > We are glad that the reviewer’s misunderstanding regarding the proofs of Theorem 7, Theorem 2, and Lemma 6 has been clarified.
> > > > >
> > > > > We also appreciated that the reviewer recognized that the experiments suggested in the previous comments (i.e., comparison with GAT) were already included in our initial submission.
> > > > >
> > > > > We would again like to thank the reviewer for the comments. We make the following comments in response to the final set of issues raised by the reviewer.
> > > > >
> > > > > ## On parameter sharing.
> > > > >
> > > > > Please note Eq 6. The message passing on the neural tree does implement parameter sharing. Contrary to what the Reviewer notes in the summary, we do not propose a “non-parameter sharing design.” Regarding Eq 8, we have already pointed out - and the reviewer agreed - that it can be implemented by Eq 6 (see the first line under “On parameter sharing” https://openreview.net/forum?id=UwSwML5iJkp&noteId=tvUVyr7hpa6).
> > > > >
> > > > > We also agree with the reviewer that conceptually the hidden dimension needed for this may be high. However, in practice, we are able to outperform traditional GNNs with a reasonable choice of hidden dimension (see Table 1 in Sec. 7).
> > > > >
> > > > >
> > > > > ## On the discrepancy between theory and experiments.
> > > > >
> > > > > As mentioned in the previous rebuttal, our theoretical results focus on the number of parameters in the network, rather than the hidden dimension, hence we believe there is no conflict with how we tune the hidden dimension in our experiments.
> > > > >
> > > > > Regarding the comment about overfitting: since the experiments suggest otherwise, we believe this criticism to be unfounded. We see the fact that Neural Tree outperforms existing methods while using a small hidden dimension to be an advantage, not a limitation of the proposed approach.
> > > > >
> > > > >
> > > > > ## On scalability.
> > > > >
> > > > > We would like to re-iterate here that the minimum complexity of approximating a G-compatible function grows exponential in the treewidth of the graph (see ref. [11]). The neural tree architecture is able to approximate any G-compatible function with the same order of parameters. Therefore, the reviewers’ concern regarding scalability seem unfounded.
> > > > >
> > > > > In the final version, we shall highlight this fact more so as that such confusions do not linger in a reader.
> > > > >
> > > > > ## On relation with GNN.
> > > > >
> > > > > We believe this is an observation more than a concern, but it is worth noting we essentially apply a standard GNN message passing on a “transformed” graph, so the connection with GNNs is quite strong.
> > > > >
> > > > > ## On 3D Scene Graphs and Experimental Results.
> > > > >
> > > > > We are actively applying the neural tree architecture to perform node labeling and prediction on 3D scene graphs and we observe significant performance gains in comparison to traditional GNN (see Table 1 in Sec. 7.1).
> > > > > Performing inference on 3D scene graphs was also the primary problem that we were investigating and that led us to the neural tree architecture.
> > > > >
> > > > > We have not claimed scalability of the proposed architecture based solely on the 3D scene graph experimental results.
> > > > >
> > > > > ## On GAT Results.
> > > > >
> > > > > In Figure A2, in the supplementary material, we observe that the neural tree outperforms message passing on input graph on PubMed citation network that has close to 20,000 nodes and 44,000 edges. This is true when we use GAT message passing (see Figure A2(d)).
> > > > >
> > > > > Therefore, the reviewers' comment that the neural tree outperforms traditional GNNs only on 3D scene graphs, and that it would be at a disadvantage on either larger graphs and/or in comparison to GAT is not true.
> > > > >
> > > > > # Summary.
> > > > >
> > > > > We thank again the reviewer for the assessment of the paper. We also appreciated the fact that the reviewer liked the originality of the design, recognized the method is reasonable, mentioned the method enjoys nice theoretical properties and presents novel insights, and suggested the paper is well-written. The reviewer also suggested this paper can trigger future exploration. This is in addition to the fact that we show a substantial performance boost in relevant applications.
> > > > >
> > > > > We wonder how a paper with these characteristics can be so far from the acceptance bar. We agree with the reviewer that adding more results (e.g. bound on hidden dimension) is always desirable, but we wonder if that justifies a “reject” rating also considering this paper has almost 20 pages of supplementary material containing complete proofs and experimental results.

---

> > > > > > ### Comment · Reviewer_zPgP · 2021-09-01
> > > > > > **Parameter Sharing**
> > > > > >
> > > > > > Thanks for your quick responses.
> > > > > >
> > > > > > We have discussed the relation between Eq 6 and parameter sharing in previous posts, as you pointed out (https://openreview.net/forum?id=UwSwML5iJkp&noteId=tvUVyr7hpa6). However, as I also mentioned, if the dimension needs to grow with the graph size, this doesn't seem to me a true parameter sharing scheme -- this is just artificially fitting a non-parameter sharing design into a parameter sharing equation. According to your argument, if we can set the hidden dimension to be arbitrarily large, anything can be expressed in a parameter sharing way. We can just set the hidden dimension size to be the dataset size.
> > > > > >
> > > > > > Your theorems also clearly suggest that the required number of parameters grows with the graph size.
> > > > > >
> > > > > > To keep it simple, can you show me how can a GCN / GraphSAGE aggregation on your neural tree implement line 248, Appendix? What is a realizable weight matrix of GCN / GraphSAGE that can disable parent-child propagation while keep child-parent message flow?

---

> > > > > > > ### Author Response · Authors · 2021-09-01
> > > > > > > **Reply**
> > > > > > >
> > > > > > > We can definitely implement GCN, GraphSAGE, or any other architecture with Neural Tree by just substituting the appropriate aggregation function $AGG_t$ in Eq (6).
> > > > > > >
> > > > > > > However, we believe the reviewer is asking whether we can make the proof tighter and obtain bounds on the hidden dimension by removing the dependence on the number of nodes in the graph. Currently, we cannot do that (and it is not even clear if one can indeed obtain a constant bound on the hidden dimension), but we feel that this should not detract from the result in the paper, which is already the first of its kind.
> > > > > > >
> > > > > > > We expect that our results can be refined to better explain the empirical evidence that significant gain in prediction accuracy is possible for a small and reasonable choice of hidden dimension (definitely not of the size of the datasets!). However, we still struggle to see how this can be the rationale behind such a negative evaluation and what we can do to fix that.
> > > > > > >
> > > > > > > ### Food for thought:
> > > > > > >
> > > > > > > * What in your view constitutes a *true* parameter sharing scheme?
> > > > > > >
> > > > > > > * Do you suggest that a traditional GNN, with local message passing, can approximate any function defined on an input graph? And do so with a bound on the hidden dimension that is constant in the graph size? Do you think this is possible?

---

> > > > > > > > ### Comment · Reviewer_zPgP · 2021-09-02
> > > > > > > > **Response**
> > > > > > > >
> > > > > > > > It's a bit frustrating to see the discussion driven a bit off-topic. I am struggling to see how I can articulate my questions more clearly. I would definitely suggest the authors read my questions carefully before posting response.
> > > > > > > >
> > > > > > > > My question is simple, if you claim that GCN or GraphSAGE can implement the Neural Tree propagation steps (e.g., (1) A21, (2) A22, (3) line 248, Appendix), just show me what would the weight matrix look like. I have never asked for any tighter bound or any more theoretical statement -- I well understand you have enough theorems so additional theorems would be too much for a single paper. However, please just show me one example GNN architecture / weight to realize the above three propagation patterns. I don't think this is too much to ask for.
> > > > > > > >
> > > > > > > > If you want more context: for GCN, the propagation performs $\sigma(AXW)$, meaning it will aggregate all neighbors (i.e., including parent and children) no matter what the weight matrix $W$ is. However, line 248, you need to disable the parent-child path while in other steps like A21, you need to re-enable them. I am repeating myself on this part as I have mentioned such reasoning in previous posts.
> > > > > > > >
> > > > > > > > If you want to discuss the general AGG function of Eq 6 rather than GCN or GraphSAGE, I'm fine with it too. If you need to use a large hidden dimension for your example, I am also OK with it even though this is not ideal. However, you need to show me that such implementation exists.
> > > > > > > >
> > > > > > > > Again, I have *not* questioned whether your *empirical results* can admit small dimension. I am well aware that your experiments are based on small dimensions. I am trying to understand how your theorems can justify your claims.
> > > > > > > >
> > > > > > > > For your "Food for thought":
> > > > > > > >
> > > > > > > > What do you think is "shared" for a parameter sharing design? Isn't it that we share parameters among data points? The required number of parameters is linear with the graph size. Then what is shared?
> > > > > > > >
> > > > > > > > As I said in previous posts, it is not the form of Eq 6 that makes the design "parameter sharing". As an analogy, for CNNs on images, when we call CNNs as being parameter sharing, is it because of
> > > > > > > >
> > > > > > > > A. the convolution operation, or
> > > > > > > >
> > > > > > > > B. the fact that the filter size does not grow with the image size
> > > > > > > >
> > > > > > > > I suppose the answer to the above is B. Imagine otherwise: if you still perform convolution on images, but your filter is always as large as the raw image, would you still call such design as parameter sharing?
> > > > > > > >
> > > > > > > > > Do you suggest that a traditional GNN, with local message passing, can approximate any function defined on an input graph? And do so with a bound on the hidden dimension that is constant in the graph size? Do you think this is possible?
> > > > > > > >
> > > > > > > > No. I have never suggested that. I don't know which part of my response gave you such impression. I understand the contribution of neural tree in improving the expressiveness. And I do appreciate it. I thought we are talking about parameter sharing, scalability and other perspective, rather than expressiveness.

---

> > > > > > > > > ### Author Response · Authors · 2021-09-02
> > > > > > > > > **Thanks and Summary - II**
> > > > > > > > >
> > > > > > > > > We would also like to thank the reviewer for pointing out that we do not provide the exact equation for GCN and GraphSAGE aggregation functions. In the final version, we will include the detailed implementation equation for GCN, GraphSAGE, GAT, GIN aggregation function in the supplementary material so that this confusion is averted for a reader.
> > > > > > > > >
> > > > > > > > > We thank the reviewer for re-iterating that he/she understands the "contribution of neural tree in improving the expressiveness" and clarifying that he/she is not asking for any more theoretical results.
> > > > > > > > >
> > > > > > > > > We believe other perspectives raised by the reviewer - parameter sharing, scalability - are important and we have already addressed them in our comments (see https://openreview.net/forum?id=UwSwML5iJkp&noteId=5wL74PBihY8).
> > > > > > > > >
> > > > > > > > > As noted there, message passing on the neural tree does implement parameter sharing (see Eq 6) and is scalable - as the number of required parameters is of the same order as the minimum complexity of approximating any G-compatible function.
> > > > > > > > >
> > > > > > > > > We, however, cannot see how the reviewer’s comments justify a "reject" rating for the paper. We would like to restate our summary of this review posted in (https://openreview.net/forum?id=UwSwML5iJkp&noteId=5wL74PBihY8).
> > > > > > > > >
> > > > > > > > > ### Summary.
> > > > > > > > >
> > > > > > > > > We thank again the reviewer for the assessment of the paper. We also appreciated the fact that the reviewer liked the originality of the design, recognized the method is reasonable, mentioned the method enjoys nice theoretical properties and presents novel insights, and suggested the paper is well-written. The reviewer also suggested this paper can trigger future exploration. This is in addition to the fact that we show a substantial performance boost in relevant applications.
> > > > > > > > >
> > > > > > > > > We wonder how a paper with these characteristics can be so far from the acceptance bar. We agree with the reviewer that adding more results (e.g. bound on hidden dimension) is always desirable, but we wonder if that justifies a “reject” rating also considering this paper has almost 20 pages of supplementary material containing complete proofs and experimental results.

---

> > > > > > > > > > ### Comment · Reviewer_zPgP · 2021-09-02
> > > > > > > > > > **Reviewer Summary**
> > > > > > > > > >
> > > > > > > > > > Thanks for your followup. In fact, what I am really looking for now is just one example, not the high level summary of contributions (as I believe I understand those contributions -- no need to re-iterate). As long as I see one such example, I will be convinced and this example will resolve many of my remaining concerns. Then the connection from neural tree to GNNs becomes solid.
> > > > > > > > > >
> > > > > > > > > > Hopefully you can see from my previous reasoning, that I doubt if such an example really exists (see "parent-child" and "child-parent" propagation pattern). That's why I insist on asking for it.
> > > > > > > > > >
> > > > > > > > > > I've read all your responses, including the links you pointed out. To be clear, let me summarize my concern this way:
> > > > > > > > > >
> > > > > > > > > > ## Summary
> > > > > > > > > >
> > > > > > > > > > Your theorems and proofs are all solid and I like your results on expressiveness.
> > > > > > > > > >
> > > > > > > > > > However, your theorems do not back up your claims. The moment you try to connect NT with GNNs, I see many issues emerging. From my view, NT is achieving the expressiveness by message passing. However, such message passing of NT is fundamentally different from the message passing of GNNs. See my comments https://openreview.net/forum?id=UwSwML5iJkp&noteId=Xby3hr2A6LZ
> > > > > > > > > >
> > > > > > > > > > For a NeurIPS paper, I consider four dimensions as suggested by the guideline: Originality, Quality, Clarity and Significance.
> > > > > > > > > >
> > > > > > > > > > For your paper, in short:
> > > > > > > > > >
> > > > > > > > > > Originality (+)
> > > > > > > > > >
> > > > > > > > > > Quality (+)
> > > > > > > > > >
> > > > > > > > > > Clarity (-)
> > > > > > > > > >
> > > > > > > > > > Significance (-)
> > > > > > > > > >
> > > > > > > > > > For Originality, I've explained my appreciation from my initial review.
> > > > > > > > > >
> > > > > > > > > > For Quality, after resolving my concerns on Lemma 6 and Theorem 7 (thanks to your explanation), I think all theorems are solid.
> > > > > > > > > >
> > > > > > > > > > For Clarity, as I mentioned above, I don't think all your main claims are well backed-up by theorems. I'm actually wondering why you insist in connecting NT with GNN. If you just present NT as a model different from GNNs (i.e., if you keep NT as the right-hand side of Eq 8, not going into Eq 6), then you would be able to make a much cleaner argument, and your theoretical results wouldn't be compromised (expressiveness and complexity stay the same). Your experiments would be simpler (although they will be more expensive since you couldn't do small hidden dimension). On the other hand, now you make a broader claim that NT is a generalization of GNNs. It doesn't seem true to me. Of course, for implementation, you can construct the H-tree and implement any GNN type of aggregation. However, note that such NT implementation is different from the one assumed by your theorems and cannot achieve the same level of expressiveness.
> > > > > > > > > >
> > > > > > > > > > For Significance, when comparing NT with GNNs, ultimately it is a trade-off between scalability / efficiency and expressiveness. The conclusion should be simple: NT is not scalable when comparing with GNNs; NT is more expressive than GNNs. However, you cannot claim scalability by comparing NT with exact inference and exact isomorphism problems, while claim expressiveness by comparing NT with GNNs. First, you need to admit such trade-off. Then, you need to properly discuss how the NT-way of exploiting the tradeoff is valuable. The significance should be evaluated by jointly considering expressiveness and scalability. The current form of the paper does not allow me to properly evaluate the significance.
> > > > > > > > > >
> > > > > > > > > > In summary, the originality and quality of the paper are both above the acceptance bar. But the clarity and significance are below. In my opinion, accepted papers shouldn't have major short-comings in any of the four criteria. The excellence in one criteria is a plus (e.g., to elevate the paper to spotlight or oral). However, the strength in one criteria could not simply make up for the weakness in another.
> > > > > > > > > >
> > > > > > > > > > ## Rating
> > > > > > > > > >
> > > > > > > > > > I understand that the standard can be interpreted differently. Thus, I fully respect the judgement of other reviewers, and I won't be upset if the paper is accepted. But I do also respect my own standard above. I enjoyed reading many parts of the paper, but I also have to be honest with the issues I see. I have put a lot of efforts to try to align theoretical results with your claims and empirical results. But eventually, I still fail to construct a consistent story to convince myself.

---

### Decision · Program_Chairs · 2021-09-27

**Decision:**

Accept (Poster)

**Comment:**

The paper proposes novel GNN architecture that is based on theory from probabilistic graphical models. The literature on GNNs is vast yet this approach seems to be unique and furthermore well grounded in theory of PGMs. Currently the manuscript focuses on undirected graphs, but extension to directed graphs should be straightforward. Very well written paper applying rich theory on PGMs in the elegant, practical way.